# Do Deeper Convolutional Networks Perform Better?

## Abstract

Over-parameterization is a recent topic of much interest in the machine learning community. While over-parameterized neural networks are capable of perfectly fitting (interpolating) training data, these networks often perform well on test data, thereby contradicting classical learning theory. Recent work provided an explanation for this phenomenon by introducing the double descent curve, showing that increasing model capacity past the interpolation threshold can lead to a decrease in test error. In line with this, it was recently shown empirically and theoretically that increasing neural network capacity through width leads to double descent. In this work, we analyze the effect of increasing depth on test performance. In contrast to what is observed for increasing width, we demonstrate through a variety of classification experiments on CIFAR10 and ImageNet32 using ResNets and fully-convolutional networks that test performance worsens beyond a critical depth. We posit an explanation for this phenomenon by drawing intuition from the principle of minimum norm solutions in linear networks.

## 1 Introduction

Traditional statistical learning theory argues that over-parameterized models will overfit training data and thus generalize poorly to unseen data (Hastie et al., 2001). This is explained through the bias-variance tradeoff; as model complexity increases, so will variance, and thus more complex models will generalize poorly. Modern deep learning models, however, have been able to achieve state-of-the-art test accuracy by using an increasing number of parameters (Krizhevsky et al., 2012; Simonyan & Zisserman, 2015; He et al., 2016). In fact, while over-parameterized neural networks have enough capacity to interpolate randomly labeled training data Zhang et al. (2017), in practice training often leads to interpolating solutions that generalize well.

To reconcile this apparent conflict, Belkin et al. (2019a) proposed the double descent risk curve, where beyond the interpolation threshhold, the risk decreases as model complexity increases. In neural networks, model complexity has thus far mainly been analyzed by varying network width. Indeed, in line with double descent, Yang et al. (2020); Nakkiran et al. (2020); Belkin et al. (2019a) demonstrated that increasing width beyond the interpolation threshhold while holding depth constant can decrease test loss.

However, model complexity in neural networks can also be increased through depth. In this work, we study the effect of depth on test performance while holding network width constant. In particular, we focus on analyzing the effect of increasing depth in convolutional networks. These networks form the core of state-of-the-art models used for image classification and serve as a prime example of a network with layer constraints. In this paper we answer the following question: *What is the role of depth in convolutional networks?*

In contrast to what has been shown for increasing model complexity through width, we demonstrate that test performance of convolutional networks worsens when increasing network depth beyond a critical point, suggesting that double descent does not happen through depth. Figure 1 demonstrates the difference between increasing width and depth in ResNets (He et al., 2016) trained on CIFAR10. In particular, Figure 1a shows that increasing width leads to a decrease in test error even when training accuracy is $100\%$. This effect is captured by the double descent curve. On the other hand, Figure 1b demonstrates that training ResNets of increasing depth but fixed width leads to an increase

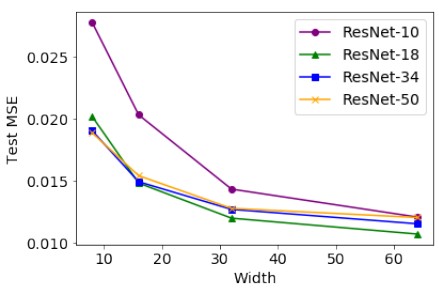 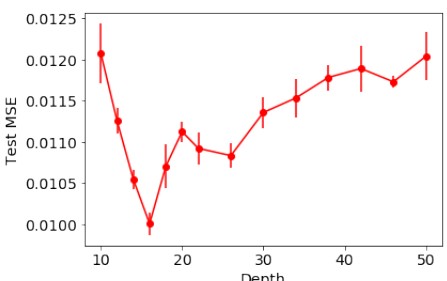

(a) ResNet-18 and ResNet-34 of increasing width on CIFAR10

(b) ResNet of width 64 on CIFAR10.

Figure 1: (a) As explained by double descent, increasing width in ResNets trained on CIFAR10 results in a decrease in test error. (b) In contrast, increasing the depth of ResNets trained on CIFAR10 results in an increase in test loss (results are averaged across 3 random seeds).

in test error. Since network depth is a form of model complexity, this behavior contradicts what is expected based on double descent. It is therefore critical to carefully analyze and understand this phenomenon.

The main contributions of our work are as follows:

1. We conduct a range of experiments in the classification setting on CIFAR10 and ImageNet32 using ResNets, fully-convolutional networks, and convolutional neural tangent kernels, and consistently demonstrate that test performance worsens beyond a critical depth (Section 3). In particular, in several settings, we observe that the test accuracy of convolutional networks is even worse than that of fully connected networks as depth increases.

2. To gain intuition for this phenomenon we analyze linear neural networks. We demonstrate that increasing depth in linear neural networks with layer constraints (e.g. convolutional networks or Toeplitz networks) leads to a decrease in the Frobenius norm and stable rank of the resulting linear operator. This implies that increasing depth leads to poor generalization, when solutions of lower Frobenius norm (e.g. solutions learned by linear fully connected networks) do not generalize (Section 4).

3. Against conventional wisdom, our findings indicate that increasing depth does not always lead to better generalization. Namely, our results provide evidence that the driving force behind the success of deep learning is not the depth of the models, but rather their width.

## 2 RELATED WORK

We begin with a discussion of recent works analyzing the role of depth in convolutional networks (CNNs). Yang et al. (2020) study the bias-variance decomposition of deep CNNs and show that as depth increases, bias decreases and variance increases. This work observes that generally the magnitude of bias is greater than that of variance, and thus overall risk decreases. However, the focus of their analysis on depth is not on the interpolating regime. In fact, they posit that it is possible for deeper networks to have increased risk. We extend their experimental methodology for training ResNets and demonstrate that, indeed, deeper networks have increased risk.

Neyshabur (2020) studied the role of convolutions, but focuses on the benefit of sparsity in weight sharing. Their work analyzed the effect of depth on fully-convolutional networks, but only considered models of two depths. Urban et al. (2017) analyzed the role of depth in student-teacher CNNs, specifically by training shallow CNNs to fit the logits of an ensemble of deep CNNs. This differs from our goal of understanding the effect of depth on CNN's trained from scratch on CIFAR10; furthermore, the ensemble of CNNs they consider only have eight convolutional layers, which is much smaller than the deep ResNets we consider in our experiments.

Xiao et al. (2018) provides initial evidence that the performance of a CNN may degrade with depth; however, it is unclear whether this phenomenon is universal across CNNs used in practice or simply

an artifact of their specific initialization designed to train deep CNNs. In fact, Xiao et al. (2020) establish that the convolutional neural tangent kernel (CNTK) solution approaches that of the neural tangent kernel (NTK) as depth increases. In our work, we analyze the generalization of the CNTK as a function of depth in Section 3.3. We show that as depth increases, test error monotonically decreases and then increases. Lastly, Xiao et al. (2018) Figure 4a and Xiao et al. (2020) Figure 2a,b provide examples of accuracy worsening with increasing depth in CNNs, but we demonstrate this phenomenon systematically across a number of settings.

Other works have aimed to understand the role of depth in CNNs by characterizing implicit regularization in over-parameterized deep CNNs. Radhakrishnan et al. (2019) characterized the inductive bias of over-parameterized autoencoders and demonstrated that with sufficient depth, these networks become locally contractive around training examples. Zhang et al. (2020) similarly studied the role of depth in autoencoders in the more restrictive setting of a single training example. Nguyen & Hein (2018) studied optimization in deep CNNs and showed that increasing depth increases representational power, while increasing width smooths the optimization landscape. While each of these works identified forms of implicit regularization which occur with depth in CNNs, they did not provide an explicit connection to generalization in CNNs used for classification, which is the focus of our work.

On the other hand, previous works studying generalization via double descent have primarily focused on over-parameterization through increasing width. In particular, Belkin et al. (2019a) and Nakkiran et al. (2020) demonstrated that double descent occurs when increasing the width of neural networks trained on MNIST (LeCun et al., 1998) and CIFAR10 respectively. Several theoretical works demonstrated double descent theoretically (Hastie et al., 2019; Belkin et al., 2019b; Mitra, 2019; Muthukumar et al., 2020; Bibas et al., 2019; Bartlett et al., 2020), but analyzed linear or shallow non-linear models with an increasing number of features. Our work performs a similar empirical analysis to Nakkiran et al. (2020), but on the impact of depth instead of width in CNNs, thereby identifying contrasting behaviors between the two different ways of increasing model complexity.

## 3 EMPIRICAL EVIDENCE IN NON-LINEAR CLASSIFIERS

We now present our main set of experiments demonstrating that the test accuracy of convolutional networks decreases when increasing depth past a critical threshold. We begin with a demonstration

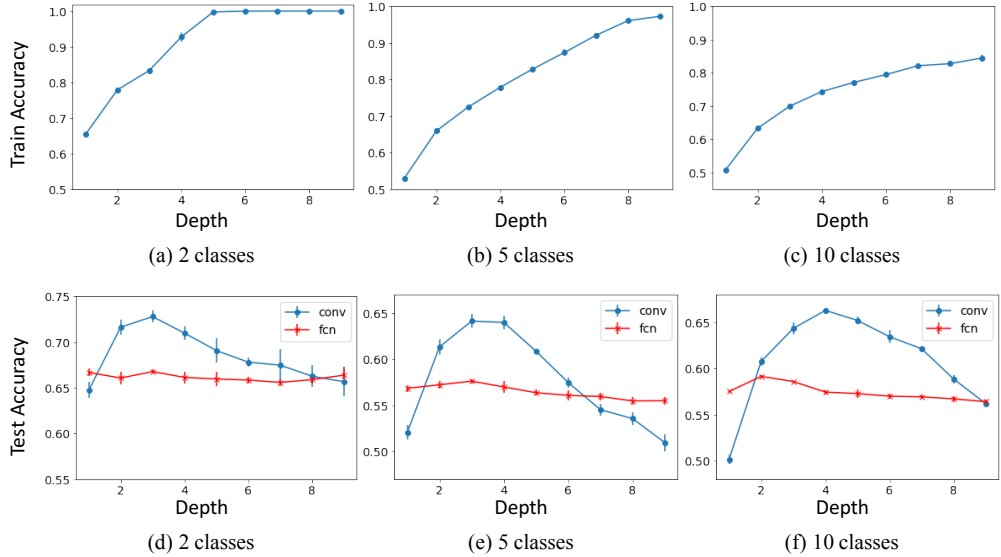

Figure 2: Train and test accuracy of the Fully-Conv Net as a function of depth, for CIFAR10 input images. Increasing depth beyond a critical value leads to a decrease in test accuracy. As depth increases, the performance of the Fully-Conv Net approaches that of a wide fully connected network (shown in red). All experiments are performed across 5 random seeds.

of this phenomenon for fully-convolutional networks applied to CIFAR10 and ImageNet32. We then demonstrate that this phenomenon holds also for ResNets applied to CIFAR10. Lastly, we show that this phenomenon occurs for the convolutional neural tangent kernel (CNTK) on subsets of CIFAR10. Our training methodology is outlined in Appendix C.

### 3.1 IMAGE CLASSIFICATION WITH FULLY-CONVOLUTIONAL NETWORKS

To understand the role of depth in convolutional networks, we begin with a simplified model of a convolutional network, which we call the *Fully-Conv Net*. The architecture of a Fully-Conv Net of depth $d$ and width $w$ for a classification problem with $c$ classes is depicted in Figure 9 of the Appendix and consists of the following layers:

- A convolutional layer with stride 1, 3 input filers, and $w$ output filters, followed by batch norm (Ioffe & Szegedy, 2015) and a LeakyReLU activation (Xu et al., 2015).
- $d - 1$ convolutional layers with stride 1, $w$ input filters, and $w$ output filters, each followed by batch norm and LeakyReLU activation.
- 1 convolutional layer with stride 1, $w$ input filters, and $c$ output filters. This is followed by an average pool of each of the output filters to produce a $c$-dimensional prediction.

Crucially, this network depends only on convolutional layers, a nonlinear activation, and batch norm; it does not depend on other components commonly found in deep learning architectures such as residual connections, dropout, downsampling, or fully connected layers. We note that this model is not designed to necessarily perform well, but rather to isolate and understand the effect of increasing the number of convolutional layers.

We trained the Fully-Conv Net on 2, 5, and 10 classes from CIFAR10 (Krizhevsky, 2009). All experiments were performed using 5 random seeds to reduce the impact of random initialization. Models were trained using Adam (Kingma & Ba, 2015) with learning rate $10^{-4}$ for 2000 epochs, and we selected the model with the best training accuracy over the course of training. We used the Cross Entropy loss, and down-sampled images to $16 \times 16$ resolution to reduce the computational burden. See Appendix C for a list of all classes used. The resulting train and test accuracies are

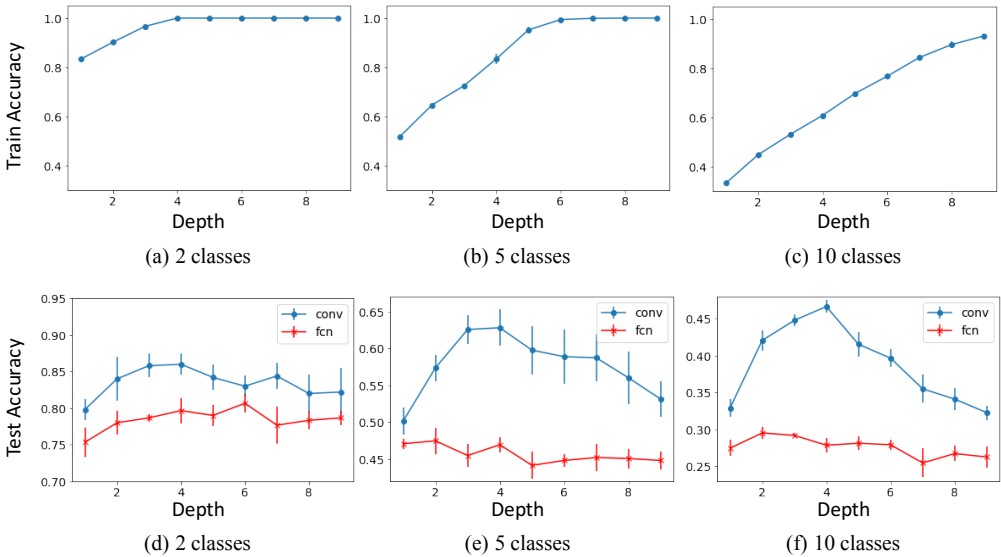

(a) 2 classes       (b) 5 classes       (c) 10 classes

(d) 2 classes       (e) 5 classes       (f) 10 classes

Figure 3: Train and test accuracy of the Fully-Conv Net as a function of depth, for ImageNet32 input images. Increasing depth beyond a critical threshold again leads to a decrease in test accuracy. Increasing depth beyond a critical value leads to a decrease in test accuracy. As depth increases, the performance of the Fully-Conv Net approaches that of a wide fully connected network (shown in red). All experiments are performed across 5 random seeds.

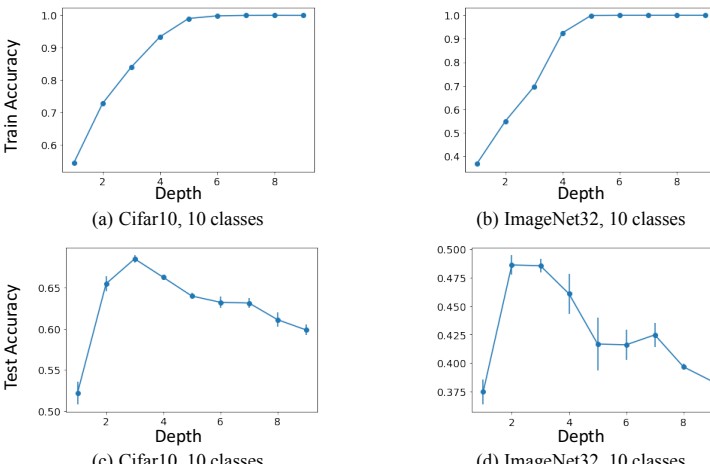

(a) Cifar10, 10 classes        (b) ImageNet32, 10 classes

(c) Cifar10, 10 classes        (d) ImageNet32, 10 classes

Figure 4: Increasing depth well past the interpolation threshold leads to a decrease in test accuracy. All experiments are across 3 random seeds. (a) We trained Fully-Conv nets of increasing depth but fixed width 32 on 10 classes of CIFAR10. (b) We trained Fully-Conv nets of increasing depth but fixed width 32 on 10 classes of ImageNet32.

shown in Figure 2. As expected, as depth increases, training accuracy becomes 100%. However, beyond a critical depth threshold, the test accuracy begins to degrade sharply. Furthermore, the value of this critical depth appears to increase as the number of training classes increases.

In addition to CIFAR10, we also applied the Fully-Conv Net to subsets of ImageNet32 (Chrabaszcz et al., 2017), which is ImageNet downsampled to size $32 \times 32$. We again trained on 2, 5, and 10 classes, using the same training procedure as for CIFAR10. Training and test accuracies for ImageNet32 are shown in Figure 3. Again, we observe that as depth increases past a critical value, test performance degrades.

**Remarks.** When training to classify between 2 and 5 classes, the test accuracy continues to decrease even when increasing depth past the interpolation threshold, i.e. even after achieving 100% training accuracy. This in contrast to double descent where increasing model complexity beyond the interpolation threshold leads to an increase in test accuracy. Interestingly, as depth increases, the test accuracy approaches that of a fully connected network. While the Fully-Conv Nets were before or at the interpolation threshold for the 10 class setting in Figures 2 and 3, Figure 4 demonstrates that a similar decrease in test accuracy occurs also after the interpolation threshold for wider models which can interpolate the data.

## 3.2 IMAGE CLASSIFICATION WITH MODERN DEEP LEARNING MODELS

To understand the effect of increasing depth in modern machine learning models, we analyzed variants of ResNet trained on CIFAR10. ResNet-18 and ResNet-34 consist of 4 stages, each of which are connected by a downsampling operation. Each stage is comprised of a number of *basic blocks*, which are two layers with a residual connection. There is a convolutional layer before the first stage, and a fully connected layer after the last stage. ResNet-18 uses 2 basic blocks in each stage, while ResNet-34 uses $(3, 4, 6, 3)$ blocks in each stage respectively. By varying the number of blocks in each stage, we constructed a variety of ResNet models of different depths; in particular, by choosing $(n_1, n_2, n_3, n_4)$ blocks in each stage, we can construct a ResNet model of depth $2 + 2 \cdot (n_1 + n_2 + n_3 + n_4)$. The width $w$ of a model is defined to be the number of filters in the first stage; there are then $(w, 2w, 4w, 8w)$ filters in each stage respectively. See Figure 10 of the Appendix for a diagram. We trained models up to depth 50, see Appendix C for a more detailed description of the models used.

For Figure 1 and Figure 5, we trained each of our ResNet models using MSE loss on a random subset of 25,000 training images from CIFAR10, and plotted the test loss as a function of depth. Our experimental methodology is based on that of Yang et al. (2020). We trained for 500 epochs

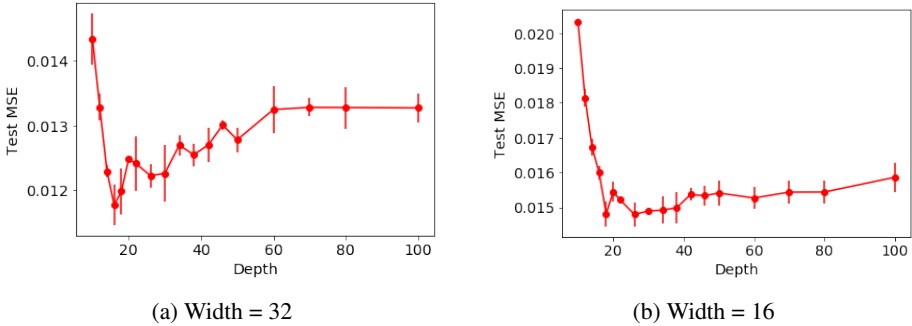

(a) Width = 32                 (b) Width = 16

Figure 5: Test loss for ResNet models of width 32 (a) and width 16 (b) increases with depth.

using SGD with learning rate 0.1 and momentum 0.9, and we decreased the learning rate by a factor of 10 every 200 epochs. We also use their data augmentation scheme of random crops and random horizontal flips on the training data. In the width 64 and 32 models, test loss increases beyond a critical depth, and in the width 16 model, test loss does not improve beyond a certain depth. Plots of the train and test losses and accuracies of all models are given in the Appendix in Figure 12.

**Remarks.** In Appendix D, we provide the training and test accuracies for the ResNets presented in this work. We also demonstrate that increasing depth in later blocks of ResNet leads to a more drastic increase in test error than increasing depth in earlier blocks.

### 3.3 IMAGE CLASSIFICATION WITH THE CONVOLUTIONAL NEURAL TANGENT KERNEL

To remove the effect of additional factors in modern neural networks such as random initialization, width, batch normalization, and down-sampling, we turn to the setting of infinite width neural networks. With proper scaling as width approaches infinity, the solution given by training neural networks is given by a solution to a corresponding kernel regression problem, where the kernel is referred to as the *neural tangent kernel*, or NTK (Jacot et al., 2018). Arora et al. (2019) computed the NTK for convolutional networks (CNTK); we use their code for the experiments in this section.

We analyze the effect of depth on generalization of the CNTK. Since we are simply running kernel regression, our predictor perfectly interpolates the training data (i.e. 100% training accuracy and 0 training error). In Figure 6, we compute the CNTK with a training set of roughly 250 examples each of CIFAR10 planes and trucks. We then calculate the test loss and accuracy on a test set of roughly 250 planes and trucks. We observe that test error is monotonically decreasing up to a critical depth, after which it is monotonically increasing. This provides further evidence that such a phenomenon occurs more generally in convolutional networks. In Appendix D.6, we present

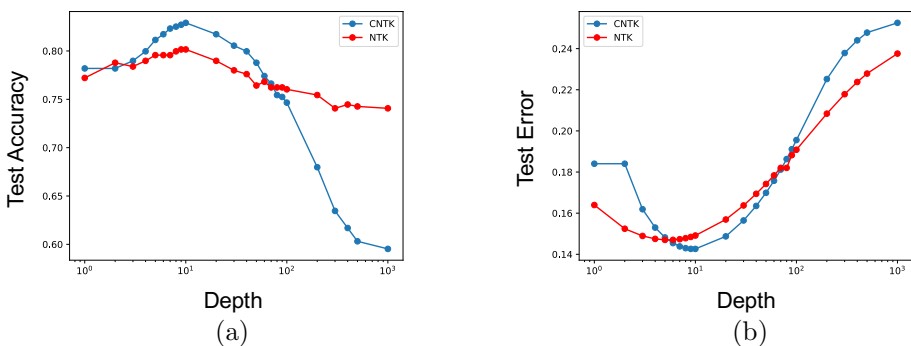

Figure 6: Test loss and accuracy for the NTK and the CNTK of increasing depths (on a log scale), trained on a subset of planes and trucks from CIFAR10. Beyond a critical depth, the test error for the CNTK increases and the accuracy decreases.

additional empirical evidence that the generalization of the CNTK worsens past a critical depth by considering classification problems with a varying number of classes.

## 4   THE ROLE OF DEPTH IN LINEAR NEURAL NETWORKS

In the previous section, we observed that past a critical depth threshold the performance of CNNs degrades with depth. For Fully-Conv networks, as depth increased, test performance appeared to approach that of a fully connected network. In this section, we aim to better understand this phenomenon, and in particular determine whether the performance of CNNs of increasing depth does indeed approach that of a fully-connected network. We turn to the setting of linear neural networks to simplify this analysis. Linear neural networks are useful for analyzing deep learning phenomena since they represent linear operators but have non-convex optimization landscapes. Furthermore, the solution learned by a single layer fully connected network is well understood as it is simply the minimum Frobenius norm solution Engl et al. (1996)[1].

In this section, we conduct experiments on linear CNNs of increasing depth in both the classification and autoencoding settings. We show that linear CNNs of increasing depth consistently produce solutions of decreasing Frobenius norm, thus approaching the solution learned by a single layer fully connected network. To connect this to generalization, we provide a specific example of a classification setting where the solution learned by a fully-connected network, and thus linear CNNs of large depth, generalizes poorly, yet shallow CNNs generalize well. We propose that a similar mechanism could also explain the decrease in test accuracy beyond a critical threshold in the non-linear setting.

### 4.1   PRELIMINARIES

Linear networks are models of the form $f(x) = W_d \cdots W_1 x$, where $W_i \in \mathbb{R}^{k_i \times k_{i-1}}$ is a weight matrix. CNNs are a special type of a constrained neural network, where each weight matrix $W_i$ belongs to a subspace $\mathcal{S}_i \subset \mathbb{R}^{k_i \times k_{i-1}}$ of dimension $r_i$. Since the product of linear operators is linear, these neural networks represent linear functions. In particular, let $W = W_d \cdots W_1$ represent the operator of a linear network. We will analyze the Frobenius norm (square root of the sum of squares of the entries) of $W$ and the stable rank (a surrogate for the rank) of $W$ in our analysis.

**Definition 1.** *(Vershynin, 2018, Chapter 7) Given a matrix $W \in \mathbb{R}^{d_1 \times d_2}$, the **stable rank** of $W$ is given by:*

$$\frac{\|W\|_F^2}{\|W\|^2} = \frac{\sum_i \sigma_i^2}{\sigma_1^2};$$

*where $\{\sigma_i\}$ denote the singular values of $W$.*

**Computing the Linear Operator Efficiently.** Given a linear neural network, it is possible to compute the linear operator by first constructing the matrix representation for each layer and then multiplying Radhakrishnan et al. (2019). Instead, we use the following simpler approach for producing the operator. Given a network implementing a map from an $s \times s$ image to $\mathbb{R}^d$, we reshape the $s^2 \times s^2$ identity matrix as an $s^2 \times s \times s$ tensor (so the batch size is $s^2$) and feed this through the network. We then reshape the output matrix to be of size $d \times s^2$ to obtain the resulting operator.

### 4.2   LINEAR CONVOLUTIONAL CLASSIFIERS

The following experiment provides an example where fully connected networks do not generalize, and in which linear convolutional classifiers of increasing depth perform similarly to a fully connected network. Consider a toy dataset of $6 \times 6$ color images as shown in Figure 7a. We use 2 training examples to represent two classes: Class label 1 has a blue pixel in the upper left hand corner and class label $-1$ has a red pixel in the upper left hand corner. We then construct 200 test examples with 100 having a red pixel and 100 having a blue pixel in a randomly selected location in the lower right $3 \times 3$ quadrant of the square.

---

[1]This assumes zero initialization for the single layer network. When the network has 1 output, the minimum Frobenius norm solution is the minimum $\ell_2$ norm solution.

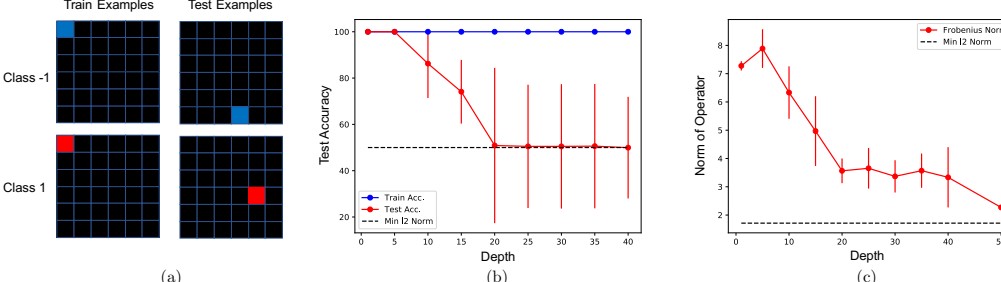

Figure 7: A toy example demonstrating that increasing depth in linear convolutional networks leads to operators of decreasing $\ell_2$ norm, which manifests as a decrease in test accuracy. (a) A visualization of samples from our toy dataset. (b) The training and test performance of linear convolutional networks of varying depth across 5 random seeds. The test accuracy of the minimum $\ell_2$ norm solution for this problem is shown as a dashed black line. (c) The $\ell_2$ norm of the operator with varying depth. The norm of the minimum $\ell_2$ norm solution for this problem is shown as a dashed black line.

While simple, this classification setting is useful for comparing the performance of CNNs and fully connected networks. It is set up to be trivial to solve using the convolution operation, but is such that the minimum Frobenius norm solution would not be able to generalize. Indeed, as demonstrated in Figure 7b, a linear fully convolutional network with 5 layers and 32 filters per layer is able to consistently get 100% test accuracy across 5 random seeds. On the other hand, performing linear regression to learn the minimum Frobenius norm solution yields a 50% test accuracy.

In Figure 7b, we see that increasing the depth of the fully convolutional network leads to a degradation in test accuracy. In particular, the average test accuracy across 5 random seeds is approximately 50% for networks of depth 20 or larger, which all obtain 100% training accuracy. In Figure 7c, we compare the Frobenius norm of the corresponding linear operator across depths and see that the Frobenius norm decreases with increasing depth. This simple example demonstrates that the performance of CNNs of increasing depth approaches that of a fully connected network, which does not generalize for many image classification tasks. In Appendix D.7, we present an additional experiment with more training samples that again demonstrates a similar phenomenon.

## 4.3 Linear Autoencoders

Similar to the case of linear convolutional classifiers, we now demonstrate that increasing depth in linear convolutional autoencoders leads to a decrease in Frobenius norm and stable rank. As done in Radhakrishnan et al. (2019); Zhang et al. (2020), we begin with the simple example of a linear convolutional autoencoder trained on a single image. When there is only a single layer, the authors in Radhakrishnan et al. (2019); Zhang et al. (2020) prove that the solution must be full rank due to the sparsity and weight sharing in a convolutional layer. This is demonstrated in Figure 8d,e where the depth 1 solution has large stable rank.

In Figure 8a,b,d,e, we demonstrate that increasing depth in linear convolutional autoencoders leads to solutions of decreasing Frobenius norm and stable rank. In particular, we observe that the norm of the trained networks decreases to approximately that of the minimum Frobenius norm solution, which is a projection onto the training examples Radhakrishnan et al. (2019); Zhang et al. (2020).

While the previous experiments have considered networks with convolutional layers, in Figure 8c, we present an example of a linear autoencoder with alternate layer constraints (Toeplitz layers) for which increasing depth again decreases the operator's Frobenius norm and stable rank. The key similarity between this layer constraint and that of a convolutional layer is that both increase representational power through depth. Indeed, the authors in Ye & Lim (2016) proved that every matrix can be decomposed as a product of Toeplitz matrices, and thus, neural networks with Toeplitz layers are expressive through depth. We note that showing that every matrix can be written as a product of convolutional layers remains open, but a simple parameter counting argument as given in Radhakrishnan et al. (2019) implies that representational power increases with depth. This experiment thus

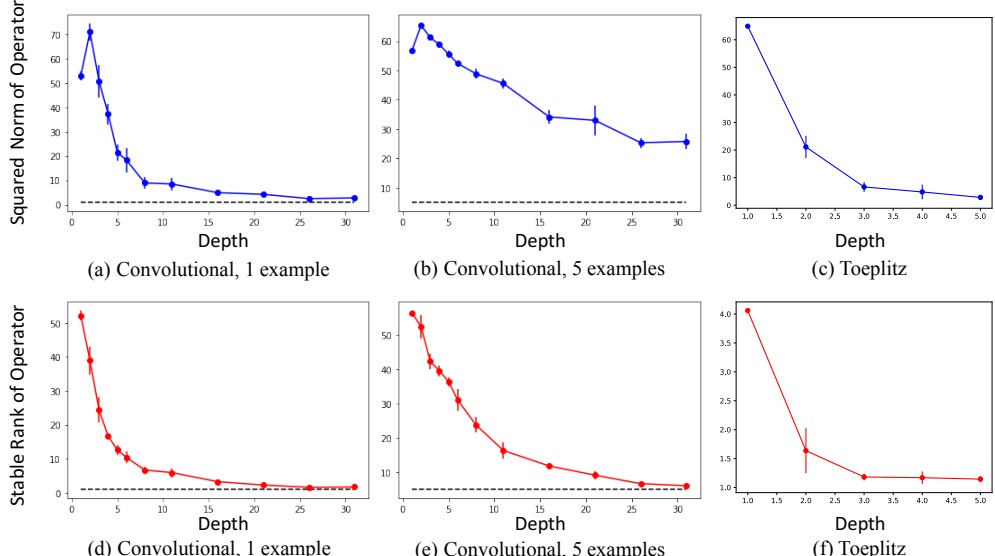

Figure 8: Training layer-constrained linear autoencoders of increasing depth leads to a decrease in the Frobenius norm and stable rank of the resulting operator. (a,d) We train convolutional networks of varying depth on 1 example. (b,e) We train convolutional networks of varying depth on 5 examples. (c,f) We train a network with Toeplitz layers to autoencode a 5 dimensional vector.

provides empirical evidence that the phenomenon of decreasing norm with increasing depth occurs generally for networks with layer constraints that are more expressive through depth.

## 5 DISCUSSION

In this work, we presented an empirical analysis of the impact of depth on generalization in CNNs. We first demonstrated that in modern non-linear CNNs, increasing depth past a critical threshold led to a decrease in test accuracy. This result is in stark contrast to the role of width in CNNs, as explained by double descent. Furthermore, for Fully-Conv Nets, we observed that increasing depth led to performance comparable to that of fully connected networks.

To better understand this phenomenon, we analyzed the operators learned by linear CNNs and demonstrated that increasing depth in these networks led to a decrease in Frobenius norm and stable rank of the learned operator. Moreover, as depth increased, we observed that the norm of the operator approached that of the minimum Frobenius norm solution, which is the solution learned by fully connected network. As demonstrated by our example in Section 4, in settings where the minimum Frobenius norm solution does not generalize, we observe that deep convolutional networks do not generalize. Understanding whether the norm of the functions learned by deep non-linear CNNs approaches that of functions learned by non-linear fully connected networks is an important direction of future work that could explain the poor generalization of deep CNNs observed in this work.

Throughout this work, we consistently observed that increasing depth in CNNs beyond the interpolation threshold led to a decrease in test accuracy. Hence, our findings imply that practitioners should decrease depth in these settings to obtain better test performance. An interesting direction for future work is to understand where the critical depth threshold occurs as a function of width and number of classes. Importantly, if, as initial evidence in this paper suggests, the critical depth is correlated with the number of classes, then practitioners should use shallow, wide convolutional networks for problems with few classes.

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

## APPENDIX

### A   FULLY CONVOLUTIONAL NETWORK ARCHITECTURE

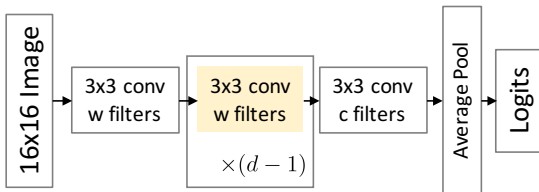

Figure 9: A diagram of the Fully-Conv Net used with width $w$, depth $d$, and $c$ classes. Each convolutional layer (except for the last one) is followed by batch norm and LeakyReLU.

### B   RESNET ARCHITECTURE

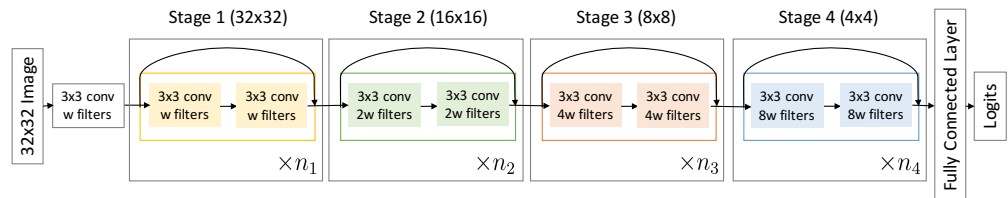

Figure 10: A diagram of the custom ResNet models used. The $i$th block is repeated $n_i$ times in stage $i$. The first convolutional layer in stages 2, 3, and 4 downsamples the input using a stride of 2, while the remaining layers have a stride of 1.

### C   EXPERIMENTAL DETAILS

All models were trained on an NVIDIA TITAN RTX GPU using the PyTorch library.

An anonymized repository with code used for this paper can be found here: `https://anonymous.4open.science/r/ebf33ffa-565e-408d-ae5f-12d91f942000/`

| Dataset | Network | Width | Activation | Optimizer | Init. | Train Size | # Epochs | Loss | Trials | Figure |
|---|---|---|---|---|---|---|---|---|---|---|
| CIFAR10 | Fully-Conv | 16 | LeakyReLU | Adam lr=1e-4 | Kaiming Normal | 5000 /class | 2000, or 100% accuracy | Cross Entropy | 5 | 2 |
| CIFAR10 | FCN | 4096 | | | | 5000 /class | | | 5 | |
| ImageNet32 | Fully-Conv | 16 | | | | 1300 /class | | | 5 | 3 |
| ImageNet32 | FCN | 4096 | | | | 1300 /class | | | 5 | |
| CIFAR10 | Fully-Conv | 32 | | | | 5000 /class | | | 3 | 4 |
| ImageNet32 | Fully-Conv | 32 | | | | 1300 /class | | | 3 | |
| CIFAR10 | ResNet | 64 | ReLU | SGD w/ momentum LR schedule from Yang et al (2020) | Default Pytorch | 2500/class | 500 | MSE | 3 | 1a, 12, 13ae |
| | ResNet | 32 | | | | | | | 3 | 5a, 12, 13bf |
| | ResNet | 16 | | | | | | | 3 | 5b, 12, 13cg |
| | ResNet | 8 | | | | | | | 3 | 13dh, 14 |
| | ResNet Increase Stage 1 | 32 | | | | | | | 1 | 15 |
| | ResNet Increase Stage 3 | 32 | | | | | | | 1 | 15 |
| | ResNet10 ResNet18 Varied kernel size | 32 | | | | | | | 1 | 17 |
| | ResNet | 32 | | | | 500/class | 2000 | | 1 | 16 |
| CIFAR10 | CNTK | Infinite | ReLU | N/A | N/A | 100/class, 250/class | N/A | MSE | 1 | 6, 18 |
| Toy classification | Linear Conv | 32 | None | Adam lr=1e-4 | Xavier Uniform | 2, 8 | Until Convergence | MSE | 5 | 7, 19 |
| Cifar10 | Linear Conv | 16 | None | Adam lr=1e-4 | Xavier Normal | 1, 5 | Until Convergence | MSE | 5 | 8 |

Figure 11: Experimental details for all experiments conducted.

| Model Depth | Blocks per Stage |
|:---:|:---:|
| 10 | 1, 1, 1, 1 |
| 12 | 1, 1, 2, 1 |
| 14 | 1, 2, 2, 1 |
| 16 | 2, 2, 2, 1 |
| 18 | 2, 2, 2, 2 |
| 20 | 2, 2, 3, 2 |
| 22 | 2, 3, 3, 2 |
| 26 | 3, 3, 3, 3 |
| 30 | 3, 4, 4, 3 |
| 34 | 3, 4, 6, 3 |
| 38 | 3, 4, 8, 3 |
| 42 | 3, 4, 10, 3 |
| 46 | 3, 4, 12, 3 |
| 50 | 3, 4, 14, 3 |

Table 1: Stage breakdown for all ResNet models used.

| # Classes | Classes used |
|:---:|:---:|
| 2 | dog, cat |
| 5 | bird, cat, deer, dog, horse |
| 10 | all classes |

Table 2: CIFAR10 classes considered in this work.

| # Classes | Classes used |
|:---:|:---:|
| 2 | kit_fox, English_setter |
| 5 | 2 classes + Siberian_husky, Australian_terrier, English_springer |
| 10 | 5 classes + Egyptian_cat, Persian_cat, malamute, Great_Dane, Walker_hound |

Table 3: ImageNet32 classes considered in this work.

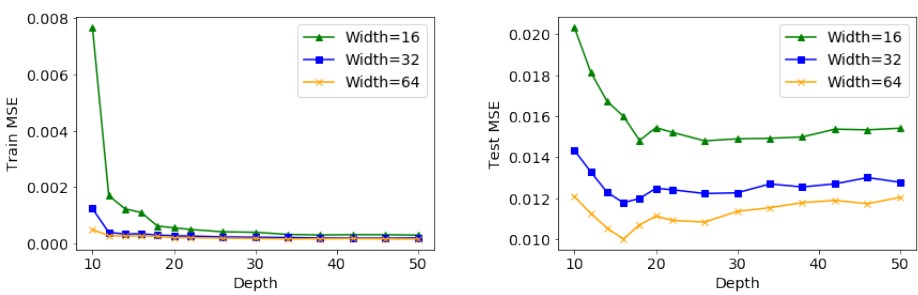

(a) Train Losses of ResNet models with widths 16, 32, and 64.

(b) Test Losses of ResNet models with widths 16, 32, and 64.

Figure 12: Train and Test losses of the ResNet models for all widths.

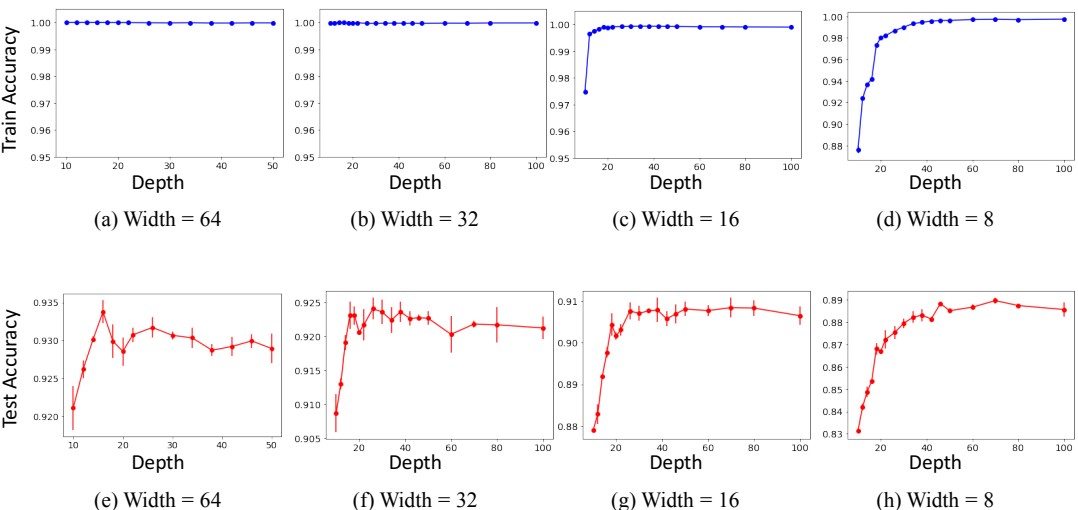

(a) Width = 64     (b) Width = 32     (c) Width = 16     (d) Width = 8

(e) Width = 64     (f) Width = 32     (g) Width = 16     (h) Width = 8

Figure 13: Train and Test accuracies for the ResNet models of width 8, 16, 32, and 64, for increasing depths.

## D    ADDITIONAL EXPERIMENTS

### D.1    ADDITIONAL RESNET PLOTS

In Figure 12, we plot the train and test losses of all ResNet models used (for widths 16, 32, 64). Additionally, in Figure 13 we plot the accuracies of all ResNet models.

### D.2    EFFECT OF DEPTH IN MODELS WITH SMALL WIDTHS

The only model in which test loss continues to increase is the width 8 model. We argue this is because the width 8 model is not sufficiently over-parameterized; in fact, in Figure 14, we see that the width 8 model is unable to reach zero training loss, while all the other models are after sufficient depth.

### D.3    EFFECT OF DOWNSAMPLING

In Figure 15 we compare a ResNet model where we increase the number of blocks in the first stage versus a model where we increase the number of blocks in the third stage. We observe that the model where the third stage blocks are increased performs worse. This is likely because adding a block in a later stage, after downsampling, increases the effective depth of the model more than adding a block in an earlier stage.

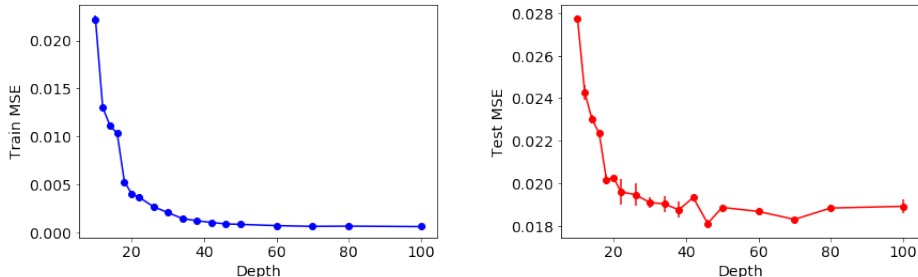

(a) Train losses for the width 8 ResNet model, for increasing depths.

(b) Test losses for the width 8 ResNet model, for increasing depths.

Figure 14: Train and test losses for the width 8 ResNet model. We see that test loss decreases as model depth increases, but train loss has still not reached 0, even for large depths.

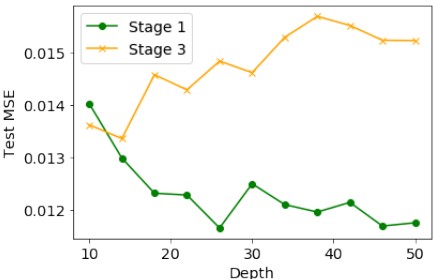

Figure 15: Test losses for the width 32 ResNet model where the 1st stage blocks are increased versus the model where the 3rd stage blocks are increased.

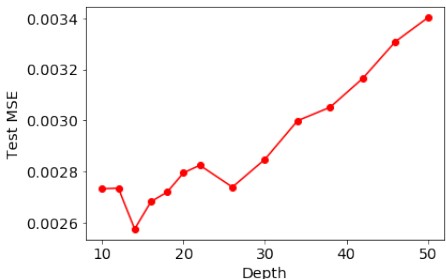

Figure 16: Test losses for the width 32 ResNet samples, trained on 500 samples per class.

### D.4 EFFECT OF NUMBER OF SAMPLES

In Figure 16 is a plot of test losses when training the width 32 ResNet model on 500 samples per class ($\frac{1}{10}$ of CIFAR10). Number of training epochs is increased accordingly. We observe that test loss increases as depth increases, showing that this phenomenon is robust to change in sample size.

### D.5 EFFECT OF KERNEL SIZE

Another form of overparameterization is increasing the kernel size for convolutional filters. In Figure 17, we train ResNet-10 and ResNet-18 of width 32 and varying kernel sizes, and observe that as kernel size increases, test loss increases. This is consistent with our proposed explanation based on expressivity, since increasing kernel size increases representational power independent of depth.

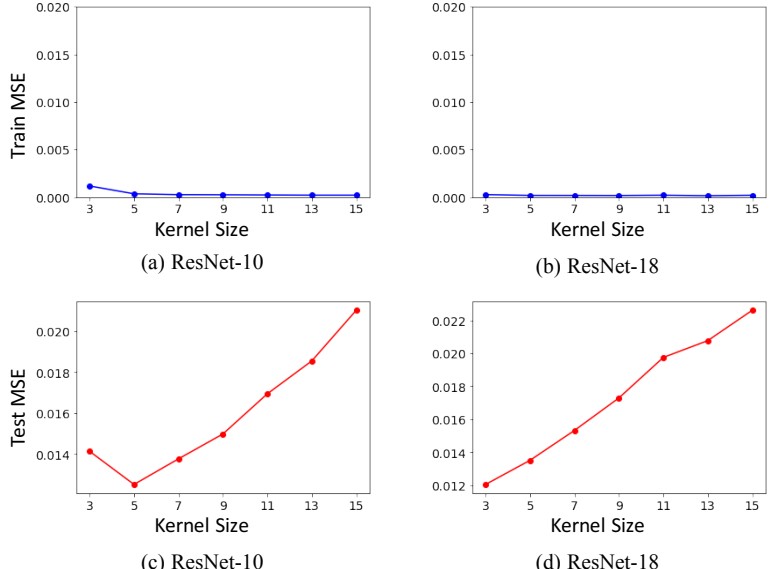

Figure 17: Train and test losses for width 32 models for increasing kernel size. We observe that test loss increases as kernel size increases.

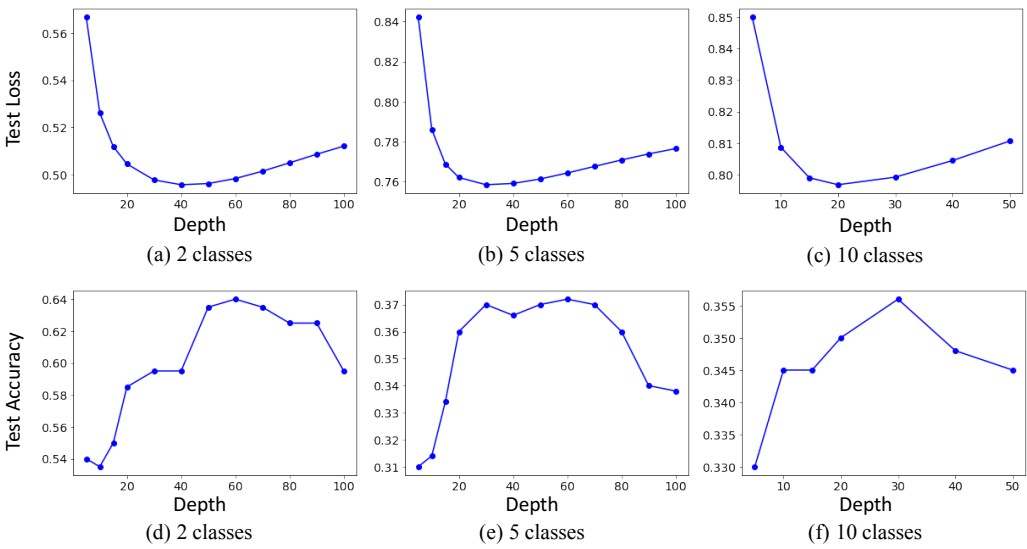

Figure 18: Test loss and test accuracy for the CNTK trained on subsets of Cifar10 with 2, 5 and 10 classes. We observe that generalization improves up to a critical depth, after which it worsens.

## D.6 ADDITIONAL CNTK EXPERIMENTS

We also train the CNTK on subsets of CIFAR10 of varying number of classes. We use 100 train and 100 test examples per class, and train on both 2 classes (birds and deer), 5 classes (cats, dogs, horses, bids, and deer), and 10 classes (all of CIFAR10). The test losses and accuracies are shown in Figure 18. Again, we see that generalization is unimodel, with test loss decreasing until a critical depth and increasing afterwards, which is in agreement with our main CNTK experiment in Figure 6.

We note that training the CNTK for large depths is computationally prohibitive. The runtime scales quadratically in the number of training samples; furthermore, training the depth 500 CNTK on 1 GPU for 500 train and 500 test samples took approximately 2 days.

## D.7 Additional Linear Convolutional Network Experiments

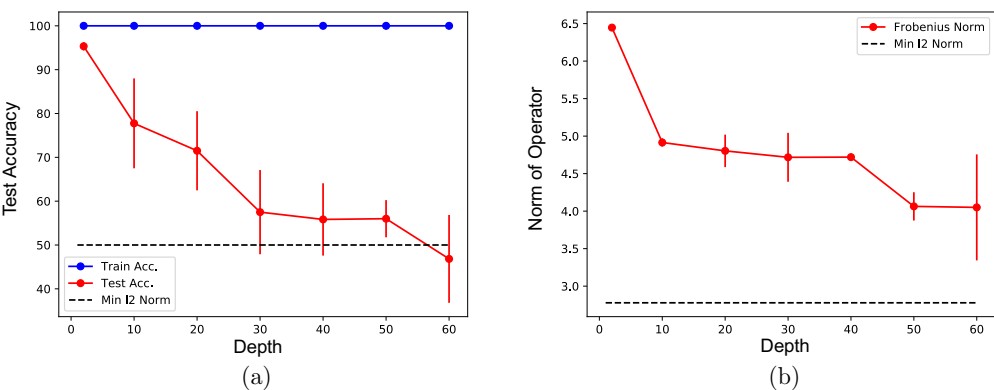

(a) (b)

Figure 19: An additional toy example demonstrating that increasing depth in linear convolutional networks leads to operators of decreasing $\ell_2$ norm, which manifests as a decrease in test accuracy. Instead of having only 1 training sample of each class, we now sample 4 from each class randomly from the upper left quadrant of a $6 \times 6$ square. Our network uses 64 filters per layer, with a kernel size of 3, and zero padding. (a) The training and test performance of linear convolutional networks of varying depth across 3 random seeds. The test accuracy of the minimum $\ell_2$ norm solution for this problem is shown as a dashed black line. (c) The $\ell_2$ norm of the operator with varying depth. The norm of the minimum $\ell_2$ norm solution for this problem is shown as a dashed black line.

## D.8 Test losses for Fully Convolutional Experiments

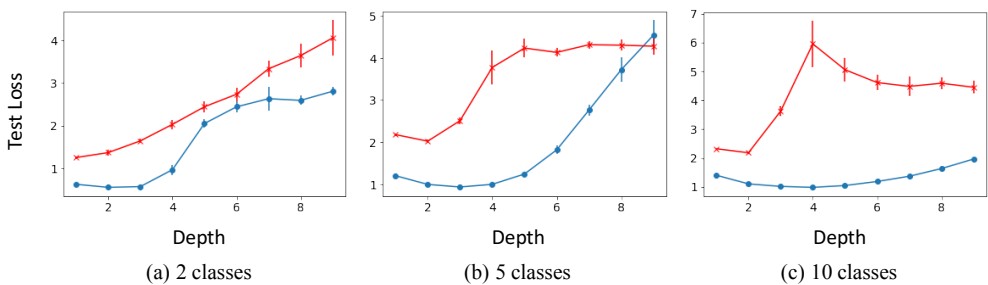

(a) 2 classes (b) 5 classes (c) 10 classes

Figure 20: Test losses for models in Figure 2 (on CIFAR10). The error for the convolutional networks is in blue and that of full connected networks is in red.

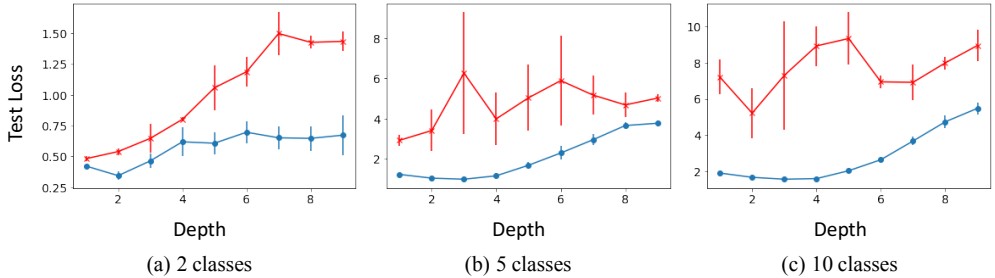

(a) 2 classes  (b) 5 classes  (c) 10 classes

Figure 21: Test losses for models in Figure 3 (on ImageNet32). The error for the convolutional networks is in blue and that of full connected networks is in red.

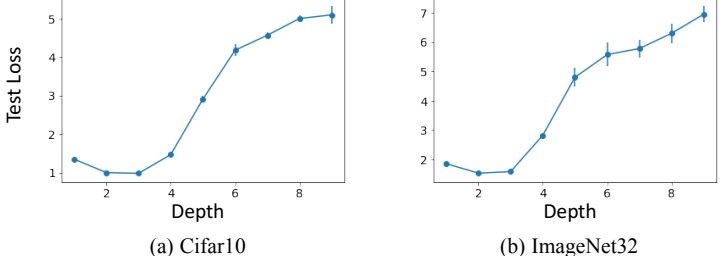

(a) Cifar10  (b) ImageNet32

Figure 22: Test losses for the models in Figure 4.

