# OpenReview forum: "Do Deeper Convolutional Networks Perform Better?"
_ICLR.cc/2021/Conference — Reject_

### Official Review · AnonReviewer4 · 2020-10-28
**A very interesting problem, but vague experimental results and explanations**

**Rating:** 5
**Confidence:** 4

**Review:**

### Post-rebuttal update:

I am raising my score from 4 to 5, to recognize extensive updates to the paper experiments and numerous clarifications during the discussion, as well as to acknowledge that some of my initial concerns were not justified, e.g. asking for similar studies for MLPs (which was effectively done in the original submission), or questioning whether the Frobenius norm in Figure 7 really goes down (which was demonstrated better in the latest update).

However, I cannot give it a higher score / advocate for acceptance because I still find section 4 (and follow-up discussion) to be more misleading/confusing than helpful when it comes to explaining the observed phenomena.

Precisely, even after the discussion, I still believe that

1) Experiments in Figure 7 should be done with circular-padded convolutions (to rule out the more trivial explanation of the decreasing norm), and on small subsets of CIFAR10/Imagenet32 (to establish whether low-norm is indeed associated with poor performance on image classification).

2) A precise definition of norm for nonlinear networks should be provided, and a link with the Frobenius norm of the linear networks should be established. I don't understand what exactly "the norm in the corresponding RKHS" means, and how the reader was supposed to infer it from the text or our earlier discussion. Further, if possible, this norm should also be evaluated on the actual nonlinear networks, and compared to other notions of norms discussed in prior literature (e.g. https://arxiv.org/pdf/1706.08947.pdf), where lower norm is typically associated with better test accuracy.

3) A proper discussion about the non-monotonic dependence on depth in the context of provided intuition should be given. Current intuition can be interpreted as either predicting monotonic decrease in accuracy with depth (learning solutions of lower and lower norms), or as predicting a sharp drop in accuracy at a certain depth (point where the minimum-norm solution can be learned), but neither interpretation explains the hill-shaped dependence, which again makes me question whether this is indeed the right explanation.


Without section 4, the paper still has novel empirical results, but in my opinion they are neither surprising (e.g. a hill-shaped dependence of accuracy is my default expectation of any NN hyper-parameter) nor actionable (there are no hints regarding what the peak depends on / how to guess it) enough for publication at this time.

A more rigorous investigation into explaining the phenomenon, or a more comprehensive empirical exploration to identify what does and what doesn't influence the best depth, would make this a great paper at a later conference.

Best,
R4.

### Original review:

#### Paper Synopsis:

The paper empirically studies the effect of depth on generalization in CNNs, and finds that, unlike in the case of width, generalization can decrease beyond a certain critical depth. The paper proposes an intuition for why this happens, by observing that in linear CNNs, the Frobenius norm of the trained network decreases with depth, and thus potentially converges to the minimum-norm solution, which corresponds to simple linear regression. By analogy and with some empirical evidence, the paper conjectures that CNNs converge to MLPs as they become deeper.



#### Pros:

Apart from certain aspects that I discuss below, the paper is clear and easy to read. I especially appreciate considering a variety of experimental settings (different datasets, number of classes, vanilla-CNNs and ResNets), as well as toy examples. Code is provided, which is another strong point of the paper. The question asked in the title of the paper is interesting and worth investigating.



#### Cons:

My major concerns are:
1. The main takeaway message appears either trivial, or at least miscommunicated.
2. Experimental evidence is unconvincing of [the stronger, non-trivial interpretation of] the key message of the paper and the proposed intuition.
3. The proposed intuition is not convincing theoretically.

I am open to be persuaded otherwise in case if I misunderstood certain claims. Below are my specific thoughts on each of the points:
1. What concretely is the claim of the paper? Is it
	1. _“For each task, architecture, and training algorithm, there is a depth optimal for generalization?”_ If so, this is a truism.
	2. _“..., in addition, beyond this optimal depth, generalization monotonically decreases?”_ This is not supported by Figures 12.(f, g), where it remains roughly flat or even goes up, while train accuracy and loss seem to have reached the optimum per Figures 12.(b, c), and 11.(a).
	3. _“among CNNs with 100% training accuracy, the best depth for generalization is the smallest one”_ (per the main contribution #3, page 2)? This is not supported by Figure 1.(b), 12.(e, f), where the best network is not the most shallow.
	4. _“The critical depth threshold is independent of certain parameters of the task/architecture/training algorithm”_? Sadly, the paper does not answer this question, but admits this would be a good direction for future work (I agree).

  As such, my takeaway from the paper is only that _“Double descent usually doesn’t happen through depth”_ in CNNs. Unfortunately, I find this observation quite trivial, and not particularly novel, see for instance [1, Figure 3].

2. Below, I interpret the claim of the paper as _“double descent doesn’t happen in CNNs; further, CNN performance converges to that of MLPs as they become deeper”_. Firstly, Figures 2.(b, c, e, f), and 3.(c, f) are all in the classical regime (< 100% training accuracy), and therefore somewhat unrelated to the claim of the paper, which concerns generalization trends of 100% accurate networks (to be clear, including the whole range of depths from 1 is highly appreciated, but to corroborate the claims of the paper multiple depths beyond 100% training accuracy are needed). Further, no plot shows a convincing _convergence_ of CNN performance to that of an MLP, rather, I see that in some cases the curves _intersect_ (in Figure 2, but not in Figure 3). Finally, Figures 12.(f, g) showcase settings where the double descent arguably does occur (i.e. flat or increasing test performance), which makes the claim that it does not happen through depth less robust and more hyper-parameter dependent.

3. Regarding the intuition about CNN to MLP convergence. My key concern is that most image classification datasets, including both considered CIFAR-10 and ImageNet32 cannot be perfectly fit with a linear function. The linear regression solution will not interpolate the training data. Therefore, whatever mechanism is behind the degradation of performance of deep nonlinear CNNs, it cannot possibly be convergence to a linear solution, because they fit the training data perfectly, and the linear solution does not. As such, I believe the intuition that may work in the deep linear case, is _guaranteed_ to not apply to nonlinear networks. If my interpretation of the intuition is wrong, I kindly ask the authors to clarify their reasoning, explicitating precise logical steps and assumptions made.


Other, more specific suggestions that may improve the quality of the paper and make the results more convincing:
1. I believe the paper would be much stronger if it performed (or referenced, if applicable) equivalent studies, including toy examples, for deep MLPs, and compared/contrasted findings with CNNs. This would be especially useful to help validate claims about CNN to MLP convergence. For example, is the decreasing norm with depth specific only to deep linear networks with constraints, or to deep linear networks in general?
2. I find the experiment in Figure 6 very interesting, but I think it can be strengthened in a few ways. Firstly, could you please provide measurements for depths < 5, as in other figures? Secondly, I assume “SAME”, zero padding was used (if not, please let me know and ignore the following point). In this case, the norm of the deep linear CNN would decrease with depth even without training, at initialization, due to convolving over more and more zeros at the edges as one goes deeper into the network. Therefore, it would be important to control for this effect (for example by using circular-padded convolutions). Finally, I believe this same toy experiment could have been conducted on a small subset of CIFAR-10 (that can be fit linearly), and the results would be more convincing, since I am otherwise not confident the observed effect is not due to some quirk of this specific toy task (e.g. would the same trend be observed if the test samples were everywhere in the image, and not at the bottom quadrant? What if train samples were in the center? What if the training set had multiple squares? etc).


#### To summarize:
1. am convinced by the observations that double descent _does not necessarily happen_ through depth in CNNs.
2. I am not convinced that it never happens, and I am not convinced one can use it in practice to select optimal depth.
3. Further, I am not convinced with the intuition and provided evidence that deep CNNs converge to MLPs.

Unfortunately, I find (1) on its own not sufficiently surprising / useful / novel [1, Figure 3] for publication.

[1] [Dynamical Isometry and a Mean Field Theory of CNNs: How to Train 10,000-Layer Vanilla Convolutional Neural Networks, Xiao et al, 2018](https://arxiv.org/abs/1806.05393)

---

> ### Author Response · Authors · 2020-11-18
> **Response to Reviewer 4**
>
> We thank the reviewer for their detailed feedback. We have updated our paper with the following additional experiments:
>
> 1. We have added a subsection of experiments (Section 3.3, Appendix D.6) in the infinite-width regime, using the Convolutional Neural Tangent Kernel (CNTK). The CNTK is deterministic (eliminating the need for random seed sampling), and perfectly interpolates the data at all depths. We again observe that the test loss is monotonically increasing beyond a critical depth.  Our experiments additionally show that for large depths the performance of the CNTK is comparable to that of the NTK.
> 2. We have run our main ResNet experiment at depths up to 100 (Figures 5, 13 and 14), which makes clearer the trend that loss is increasing beyond a critical depth.
> 3. Per your suggestion, we ran additional experiments for the toy dataset in Section 4 and Appendix D.7.
>
> We now address your concerns below.:
> * “What concretely is the claim of the paper?”
>     * The main claim of our paper is the following: “In contrast to the case of width, double descent does not occur in convolutional neural networks of increasing depth. There is an optimal depth, after which test performance generally decreases and approaches that of fully connected neural networks.”
> * “This is not supported by Figures 12.(f, g), where it remains roughly flat or even goes up, while train accuracy and loss seem to have reached the optimum per Figures 12.(b, c), and 11.(a). [...] Finally, Figures 12.(f, g) showcase settings where the double descent arguably does occur (i.e. flat or increasing test performance), which makes the claim that it does not happen through depth less robust and more hyper-parameter dependent.”
>     * For ResNet (Figures 1b, 5), the test loss increases past a critical depth. The results are less pronounced for the test accuracy because test accuracy is only a proxy for the quantity being optimized. We have updated our paper by adding depths to our ResNet plots (Figures 5, 13 (previously 12), 14 (previously 13)), which more convincingly shows that test loss increases and test accuracy decreases past a critical depth.
> * “`among CNNs with 100% training accuracy, the best depth for generalization is the smallest one’ (per the main contribution #3, page 2)? This is not supported by Figure 1.(b), 12.(e, f), where the best network is not the most shallow.”
>      * We are not claiming that the best depth for generalization is the smallest one, but that there is a critical depth threshold past which test performance generally decreases. In fact, we have now updated the paper to include convolutional neural tangent kernel (CNTK) experiments (Section 3.3, Appendix D.6) in which all models achieve 100% training accuracy, but the test loss monotonically decreases and then increases.
> * “Firstly, Figures 2.(b, c, e, f), and 3.(c, f) are all in the classical regime (< 100% training accuracy), and therefore somewhat unrelated to the claim of the paper”
>     * This was the reason we originally included the experiments in Figure 4, which present the extension of the networks from Figures 2 and 3 to the over-parameterized regime (through increasing width).  Even in this setting, we continue to see a decrease in test accuracy as depth increases.
> * “Further, no plot shows a convincing convergence of CNN performance to that of an MLP, rather, I see that in some cases the curves intersect”
>     * We are not claiming that the solution learned by the CNN in the nonlinear setting converges to that of a nonlinear MLP, but rather that the accuracy and error of the deep CNN becomes comparable to that of an MLP.  This is an important claim since the performance of a nonlinear MLP is suboptimal for image classification.
> * “My key concern is that most image classification datasets, including both considered CIFAR-10 and ImageNet32 cannot be perfectly fit with a linear function.”
>     * To clarify, our hypothesis is that non-linear CNNs will have performance approaching that of a non-linear MLP as depth increases (as is shown in Figures 2 and 3).   While it is indeed true that a linear network cannot interpolate Cifar10, we can achieve 100% accuracy using a non-linear MLP, and it is the accuracy of these models that we hypothesize non-linear CNNs of increasing depth will approach. For instance, all MLPs shown in Figures 2 and 3 have nonlinear activations and interpolate the training data, and we observe that the accuracy of deep CNNs approaches that of deep MLPs (and is even worse in some cases).  To give intuition for this, we analyze the generalization of linear CNNs as compared to linear MLPs and demonstrate that the norm of the operator from a deep CNN approaches that of the minimum norm solution (the solution from a linear MLP).

---

> > ### Author Response · Authors · 2020-11-18
> > **Response to Reviewer 4 (cont.)**
> >
> > Regarding specific suggestions:
> > * “I believe the paper would be much stronger if it performed (or referenced, if applicable) equivalent studies, including toy examples, for deep MLPs, and compared/contrasted findings with CNNs.”
> >     * We would like to clarify that we already compare with deep nonlinear MLPs in section 3 (Figures 2 & 3), and we did not observe any significant trend with increasing depth for these networks.  We have now included comparison with the NTK in Section 3.3 as well.
> > * “For example, is the decreasing norm with depth specific only to deep linear networks with constraints, or to deep linear networks in general?”
> >     * To clarify the experiments around decreasing norm with depth in linear neural networks: We identify that increasing depth leads to solutions of decreasing norm in networks with layer constraints.  We will definitely not observe this phenomenon in linear fully connected networks in standard regression settings since a single layer fully connected networks will already learn the minimum Frobenius norm solution, and under certain initializations (i.e. spectral initialization), deep linear networks are also guaranteed to learn the minimum Frobenius norm solution (see Proposition 2 from https://arxiv.org/abs/2003.06340).  This is the main reason why we did not use deep linear networks in Section 4.
> > * “I find the experiment in Figure 6 very interesting, but I think it can be strengthened in a few ways. Firstly, could you please provide measurements for depths < 5, as in other figures? [...] would the same trend be observed if the test samples were everywhere in the image, and not at the bottom quadrant? What if train samples were in the center? What if the training set had multiple squares?”
> >     * Thank you for finding our experiment interesting!  As requested, we have updated the figure to show smaller depths.  In particular, we use a network with 2 convolutional layers (the minimum such that we can have 1 channel output), and as expected, this network achieves 100% test accuracy regardless of the seed.  Per your suggestion, we have also added an additional experiment with more training examples in Appendix D.7, which again illustrates the trend of decreasing Frobenius norm and test accuracy with increasing depth.
> > * “In this case, the norm of the deep linear CNN would decrease with depth even without training, at initialization, due to convolving over more and more zeros at the edges as one goes deeper into the network.”
> >     * Our original experiment used convolutional networks with 0 padding (ensuring that the image size stayed the same throughout the layers).  While our experiments did use zero padding (as is common practice), we note that the norm of the operator at initialization would not necessarily decrease because of this.  For example, it is possible to initialize a deep linear convolutional network to be the identity even with zero padding, and as is proved in https://arxiv.org/abs/1810.10333, single layer convolutional networks will learn high rank solutions regardless of the initialization, even with zero padding.  Given that even the zero padded networks could learn a high rank solution (such as the identity function in autoencoding), the fact that training leads to solutions of lower rank with increasing depth is surprising.
> > * “Unfortunately, I find (1) on its own not sufficiently surprising / useful / novel [1, Figure 3] for publication.”
> >     * We thank the reviewer for pointing out the related work, and we have updated our related work section in the paper to discuss Xiao et al. (2018).  Figure 3 of Xiao et al. (2018) demonstrates that  deeper models perform worse, for the specific initialization scheme considered in their paper.  The paper hypothesizes that the decrease in test performance is due to “attenuation of spatially non-uniform modes,” which would be an artifact of the initialization scheme chosen. Our work shows that the decrease in test performance is a universal phenomenon in models used in practice.
> >     * While Xiao et al (2018) argues that the signal propagation for infinitely-wide, random convolutional networks approaches that of fully-connected networks, this does not give insight on the comparison of these two models after training.  We have added additional experiments in the CNTK setting (Section 3.3, Appendix D.6) to show that this phenomenon does indeed occur in the infinite-width regime after training.

---

> > > ### Comment · AnonReviewer4 · 2020-11-22
> > > **Reply 1/4**
> > >
> > > Thank you for clarifying many aspects of the work, and running new experiments. I now understand that your main claim is that generalization dependence on the network depth, from 0 to infinity, has a hill-shaped plot, i.e. peaks at some critical depth and decreases in each direction from there.
> > >
> > > This is admittedly less trivial than how I initially interpreted, but I still find that the novelty of this result relative to https://arxiv.org/pdf/1806.05393.pdf and https://arxiv.org/pdf/1912.13053.pdf (see precise comments about these works below) is mostly in evaluating these trends in more varied architectures/datasets/number of classes. While this is a useful result, I still believe the claim about convergence of CNN performance to that of MLP is not convincing, neither experimentally, nor theoretically, which is why I am still leaning to to reject the paper, although I am on the edge between 4 and 5 (to save time, posting my responses ASAP, will decide on exact score later), and again, open to increasing it further if authors explain their CNN to MLP convergence intuition in more detail.
> > >
> > >
> > > Below are my specific replies (Please see all 4 nested comments titled Reply 1/4, ... Reply 4/4):
> > >
> > >
> > > > The main claim of our paper is the following: “In contrast to the case of width, double descent does not occur in convolutional neural networks of increasing depth. There is an optimal depth, after which test performance generally decreases and approaches that of fully connected neural networks.”
> > >
> > > Thank you for clarifying.
> > >
> > >
> > > > For ResNet (Figures 1b, 5), the test loss increases past a critical depth. The results are less pronounced for the test accuracy because test accuracy is only a proxy for the quantity being optimized. We have updated our paper by adding depths to our ResNet plots (Figures 5, 13 (previously 12), 14 (previously 13)), which more convincingly shows that test loss increases and test accuracy decreases past a critical depth.
> > >
> > > Fair point. In this regard, could you please also show test losses for other plots (Figures 2, 3, 4, 7, 19)?
> > >
> > >
> > > > We are not claiming that the best depth for generalization is the smallest one, but that there is a critical depth threshold past which test performance generally decreases. In fact, we have now updated the paper to include convolutional neural tangent kernel (CNTK) experiments (Section 3.3, Appendix D.6) in which all models achieve 100% training accuracy, but the test loss monotonically decreases and then increases.
> > >
> > > Thank you again for clarifying! However, in this case I find the main contribution #3 on page 2 incorrect, as it recommends decreasing depth, while in reality one might be situated before or after the critical depth, hence needs to explore both directions.
> > >
> > >
> > > >This was the reason we originally included the experiments in Figure 4, which present the extension of the networks from Figures 2 and 3 to the over-parameterized regime (through increasing width). Even in this setting, we continue to see a decrease in test accuracy as depth increases.
> > >
> > > Thank you for clarifying. On a somewhat minor note, I still believe that in this case these lines should be plotted directly in Figures 2 (c, f) and 3 (c, f), instead of the underparameterized blue lines that are there currently, and plots in 2 (b, e) should also be extended for several depths values beyond the current maximum, to put all the claims and results of the paper in a shared context of interpolation. But  after your clarification, I see that your claim does not only concern the interpolation regime, but the whole range, and so these plots are admittedly also relevant.
> > >
> > > ...

---

> > > > ### Comment · AnonReviewer4 · 2020-11-22
> > > > **Reply 2/4**
> > > >
> > > > ...
> > > >
> > > > > We are not claiming that the solution learned by the CNN in the nonlinear setting converges to that of a nonlinear MLP, but rather that the accuracy and error of the deep CNN becomes comparable to that of an MLP. This is an important claim since the performance of a nonlinear MLP is suboptimal for image classification.
> > > >
> > > > > To clarify, our hypothesis is that non-linear CNNs will have performance approaching that of a non-linear MLP as depth increases (as is shown in Figures 2 and 3). While it is indeed true that a linear network cannot interpolate Cifar10, we can achieve 100% accuracy using a non-linear MLP, and it is the accuracy of these models that we hypothesize non-linear CNNs of increasing depth will approach. For instance, all MLPs shown in Figures 2 and 3 have nonlinear activations and interpolate the training data, and we observe that the accuracy of deep CNNs approaches that of deep MLPs (and is even worse in some cases). To give intuition for this, we analyze the generalization of linear CNNs as compared to linear MLPs and demonstrate that the norm of the operator from a deep CNN approaches that of the minimum norm solution (the solution from a linear MLP).
> > > >
> > > > Thank you for clarifying. However, I still find this claim and related intuition to be a weakness of the paper, and this is the main reason I am leaning to reject the paper. Precisely:
> > > >
> > > > 1. "Becomes comparable" is a vague statement, and the plots arguably demonstrate that it cannot be strengthened by much. No plot, including the new CNTK results, shows a _convergence_ of nonlinear CNN accuracy to that of the MLP accuracy; instead, I see that in some cases curves intersect (Figure 2; making me think that perhaps for larger depth CNNs can become in fact significantly worse than MLP, and not comparable), and in some cases they don't intersect (Figure 3, 6; making me think that they may never intersect, and hence CNN accuracy is still significantly better than that of an MLP). Therefore I can be convinced of the claim that accuracy of CNNs drops with depth, and can become even worse than that of an MLP, and that this trend is in contrast to how MLP accuracy evolves with depth. But I am not convinced of the current claim in main contribution #1 that "test accuracy of convolutional networks approaches that of fully connected networks as depth increases".
> > > >
> > > > 2. Discussion in section 4 definitely implies convergence of solutions, e.g. citing: "In this section, we aim to better understand this phenomenon, and in particular determine whether the solution learned by CNNs of increasing depth does indeed approach the solution learned by a fully-connected network", and " We show that linear CNNs of increasing depth consistently produce solutions of decreasing Frobenius norm, thus approaching the solution learned by a single layer fully connected network", so I would prefer the wording around what exactly you're conjecturing and how exactly you corroborate your conjecture to be more precise.
> > > >
> > > > 3. If, per your clarification, you do not claim that CNN solution converges to the MLP solution, then how else can the metrics of test accuracy and Frobenius norm be connected? In this case I don't see how Frobenius norm results give any intuition. Prior to your clarification, I understood the intuition as follows: "Deep nonlinear CNN accuracy approaches the accuracy of MLPs. We conjecture it's because they converge to similar solutions. Here's evidence of this happening in deep linear case: <section 4>". But if you do not conjecture that they converge to the same solution, then how are Frobenius norm measurement related to accuracies?
> > > >
> > > > > We would like to clarify that we already compare with deep nonlinear MLPs in section 3 (Figures 2 & 3), and we did not observe any significant trend with increasing depth for these networks. We have now included comparison with the NTK in Section 3.3 as well.
> > > >
> > > > Thank you, indeed.
> > > >
> > > >
> > > > > To clarify the experiments around decreasing norm with depth in linear neural networks: We identify that increasing depth leads to solutions of decreasing norm in networks with layer constraints. We will definitely not observe this phenomenon in linear fully connected networks in standard regression settings since a single layer fully connected networks will already learn the minimum Frobenius norm solution, and under certain initializations (i.e. spectral initialization), deep linear networks are also guaranteed to learn the minimum Frobenius norm solution (see Proposition 2 from https://arxiv.org/abs/2003.06340). This is the main reason why we did not use deep linear networks in Section 4.
> > > >
> > > > Thank you, good point.
> > > >
> > > > ...

---

> > > > > ### Comment · AnonReviewer4 · 2020-11-22
> > > > > **Reply 3/4**
> > > > >
> > > > > ...
> > > > >
> > > > > > Thank you for finding our experiment interesting! As requested, we have updated the figure to show smaller depths. In particular, we use a network with 2 convolutional layers (the minimum such that we can have 1 channel output), and as expected, this network achieves 100% test accuracy regardless of the seed. Per your suggestion, we have also added an additional experiment with more training examples in Appendix D.7, which again illustrates the trend of decreasing Frobenius norm and test accuracy with increasing depth.
> > > > >
> > > > > Thank you for extending the experiments. However,
> > > > > - I would still prefer a similar experiment to be run on a small subset of CIFAR10 to be evaluated in a less artificial setting.
> > > > > - Similarly to CNN vs MLP plots, in both Figures 7 and 19, I am not fully convinced the norm of the operator really converges to the minimum norm, instead of, for example, plateauing, which is arguably happening in Figure 7.c.
> > > > >
> > > > >
> > > > > > Our original experiment used convolutional networks with 0 padding (ensuring that the image size stayed the same throughout the layers). While our experiments did use zero padding (as is common practice), we note that the norm of the operator at initialization would not necessarily decrease because of this. For example, it is possible to initialize a deep linear convolutional network to be the identity even with zero padding, and as is proved in https://arxiv.org/abs/1810.10333, single layer convolutional networks will learn high rank solutions regardless of the initialization, even with zero padding. Given that even the zero padded networks could learn a high rank solution (such as the identity function in autoencoding), the fact that training leads to solutions of lower rank with increasing depth is surprising.
> > > > >
> > > > > To be honest I am not well familiar with the linked paper so may not have understood your point completely. Firstly, please note that my question was regarding Figure 7, not the autoencoding experiment. Secondly, while it is certainly possible to initialize networks in many different ways, and therefore it could have different Frobenius norm values at initialization, I believe in this case it is easy to measure and have a precise understanding of whether the initial norm was impacted or not, for example:
> > > > >
> > > > > ```
> > > > > import jax, jax.numpy as np, jax.random as random, jax.experimental.stax as stax
> > > > >
> > > > >
> > > > > def cnn(depth):
> > > > >   return stax.serial(*(stax.Conv(32, (3, 3), padding='SAME') for i in range(depth)))
> > > > >
> > > > >
> > > > > def norm(nn):
> > > > >   init_fn, apply_fn = nn
> > > > >   _, params = init_fn(random.PRNGKey(1), (1, 6, 6, 3))
> > > > >   inputs = np.eye(6*6*3).reshape((6*6*3, 6, 6, 3))
> > > > >   out = apply_fn(params, inputs)
> > > > >   return np.linalg.norm(out)
> > > > >
> > > > >
> > > > > for depth in range(1, 101, 10):
> > > > >   print(f'depth: {depth}, norm: {norm(cnn(depth))}')
> > > > > ```
> > > > > Gives
> > > > > ```
> > > > > depth: 1, norm: 12.448296546936035
> > > > > depth: 11, norm: 6.420310020446777
> > > > > depth: 21, norm: 3.4354686737060547
> > > > > depth: 31, norm: 1.6067434549331665
> > > > > depth: 41, norm: 0.8700923919677734
> > > > > depth: 51, norm: 0.4324045181274414
> > > > > depth: 61, norm: 0.20737676322460175
> > > > > depth: 71, norm: 0.10605859756469727
> > > > > depth: 81, norm: 0.05596714839339256
> > > > > depth: 91, norm: 0.028174329549074173
> > > > > ``````
> > > > > While I believe that the norm can change during training, I am not convinced that the resulting norm _does not depend at all_ on the initial norm. I may be missing something here though, so please correct me if so.
> > > > >
> > > > >
> > > > > > We thank the reviewer for pointing out the related work, and we have updated our related work section in the paper to discuss Xiao et al. (2018). Figure 3 of Xiao et al. (2018) demonstrates that deeper models perform worse, for the specific initialization scheme considered in their paper. The paper hypothesizes that the decrease in test performance is due to “attenuation of spatially non-uniform modes,” which would be an artifact of the initialization scheme chosen. Our work shows that the decrease in test performance is a universal phenomenon in models used in practice.
> > > > >
> > > > > Thank you for pointing this out, indeed that Figure was done with their special initialization. However, note that Figure 4.a, GS4 shows the same trend for regular normally-initialized networks, and should probably be also discussed.
> > > > >
> > > > > ...

---

> > > > > > ### Comment · AnonReviewer4 · 2020-11-22
> > > > > > **Reply 4/4**
> > > > > >
> > > > > > ...
> > > > > >
> > > > > > > While Xiao et al (2018) argues that the signal propagation for infinitely-wide, random convolutional networks approaches that of fully-connected networks, this does not give insight on the comparison of these two models after training. We have added additional experiments in the CNTK setting (Section 3.3, Appendix D.6) to show that this phenomenon does indeed occur in the infinite-width regime after training.
> > > > > >
> > > > > > Thank you for adding this new experiment. A few comments on it:
> > > > > > 1. What exactly does "For the NTK, we applied sigmoid activations prior to computing the test error so that the error remained on the same scale as the CNTK error" mean, and how is it justified? Shouldn't both models be evaluated in the same setting, to have a fair comparison?
> > > > > > 2. Could you please double-check the NTK performance numbers? Namely, it appears for many settings to fluctuate around random chance of 50%, which seems a bit low. In an attempt to reproduce it I wrote the following code:
> > > > > >
> > > > > > ```
> > > > > > !pip install neural-tangents tensorflow_datasets
> > > > > >
> > > > > > from jax import numpy as np
> > > > > > import neural_tangents as nt
> > > > > > import tensorflow_datasets as tfds
> > > > > >
> > > > > >
> > > > > > ds_train = tfds.as_numpy(tfds.load('cifar10', batch_size=-1, split='train'))
> > > > > > x_train, y_train = ds_train['image'], ds_train['label']
> > > > > >
> > > > > > planes_x = x_train[y_train == 0]
> > > > > > planes_y = y_train[y_train == 0]
> > > > > > trucks_x = x_train[y_train == 9]
> > > > > > trucks_y = y_train[y_train == 9]
> > > > > >
> > > > > > x_train = np.concatenate([planes_x[:250], trucks_x[:250]])
> > > > > > y_train = np.concatenate([np.ones((250, 1)), -np.ones((250, 1))])
> > > > > > x_test = np.concatenate([planes_x[250:500], trucks_x[250:500]])
> > > > > > y_test = np.concatenate([np.ones((250, 1)), -np.ones((250, 1))])
> > > > > >
> > > > > > for depth in [0, 1, 5, 10, 20, 30, 40, 50, 60, 70, 80, 90, 100, 200, 300]:
> > > > > >   layers = [nt.stax.Flatten(), nt.stax.Dense(1, 2**0.5)]
> > > > > >   for d in range(depth):
> > > > > >     layers += [nt.stax.Relu(), nt.stax.Dense(1, 2**0.5)]
> > > > > >
> > > > > >   _, _, kernel_fn = nt.stax.serial(*layers)
> > > > > >   predict_fn = nt.predict.gradient_descent_mse_ensemble(kernel_fn, x_train, y_train)
> > > > > >   y_test_pred = predict_fn(x_test=x_test, get='ntk', t=None)
> > > > > >
> > > > > >   loss_d = np.mean((y_test_pred - y_test)**2)
> > > > > >   acc_d = np.mean(y_test * y_test_pred > 0)
> > > > > >   print(f'depth: {depth}, acc: {acc_d}, loss: {loss_d}')
> > > > > > ```
> > > > > >
> > > > > > and got
> > > > > > ```
> > > > > > depth: 0, acc: 0.612000048160553, loss: 1.6684468984603882
> > > > > > depth: 1, acc: 0.8140000104904175, loss: 0.5931008458137512
> > > > > > depth: 5, acc: 0.8360000252723694, loss: 0.5273554921150208
> > > > > > depth: 10, acc: 0.8320000171661377, loss: 0.5180454254150391
> > > > > > depth: 20, acc: 0.8340000510215759, loss: 0.5241310000419617
> > > > > > depth: 30, acc: 0.8340000510215759, loss: 0.5366725325584412
> > > > > > depth: 40, acc: 0.8320000171661377, loss: 0.5499415397644043
> > > > > > depth: 50, acc: 0.8280000686645508, loss: 0.562644898891449
> > > > > > depth: 60, acc: 0.8280000686645508, loss: 0.5744587182998657
> > > > > > depth: 70, acc: 0.8240000605583191, loss: 0.5853518843650818
> > > > > > depth: 80, acc: 0.8200000524520874, loss: 0.5953996777534485
> > > > > > depth: 90, acc: 0.8160000443458557, loss: 0.6046954393386841
> > > > > > depth: 100, acc: 0.8160000443458557, loss: 0.6133300065994263
> > > > > > depth: 200, acc: 0.8080000281333923, loss: 0.6778715252876282
> > > > > > depth: 300, acc: 0.7980000376701355, loss: 0.735858678817749
> > > > > > ```
> > > > > > Therefore I wonder if there is some kind of numerical issue that might be impairing the performance of NTK in Figure 6?
> > > > > >
> > > > > > 3. It appears that large depth behavior of the CNTK, and notably its generalization and implications for finite networks has been studied in https://arxiv.org/abs/1912.13053 (e.g. Figure 2), and I believe drawing connections to this work both in the context of this particular Figure and this paper in general would be very useful.
> > > > > >
> > > > > > END OF COMMENTS.

---

> > > > > > > ### Author Response · Authors · 2020-11-23
> > > > > > > **Thank you for the detailed feedback. Part 1/2**
> > > > > > >
> > > > > > > Thank you very much for the detailed and helpful feedback!   Thank you for sharing code for the NTK experiments.  We have now corrected our NTK experiments in the updated paper, add CNTK and NTK comparisons at depths up to 1000, and added additional plots of test losses for fully convolutional networks as requested.  We have also updated our paper to better clarify our claims on comparing the performance of a deep CNN to that of a deep MLP.  In particular, we mainly want to emphasize that the performance of a deep CNN worsens with increasing depth after a critical depth and that this performance is actually  worse than that of an MLP in several settings.
> > > > > > >
> > > > > > > We now address your specific questions below.
> > > > > > >
> > > > > > > * “Fair point. In this regard, could you please also show test losses for other plots (Figures 2, 3, 4, 7, 19)?”
> > > > > > >     * We have now updated the paper to include the test errors for Figures 2, 3, and 4.  Unfortunately, we didn’t log the test error for figures 7 and 19 when generating these figures, but are working now on generating these and can include them in the supplementary.
> > > > > > > * “However, in this case I find the main contribution #3 on page 2 incorrect, as it recommends decreasing depth, while in reality one might be situated before or after the critical depth, hence needs to explore both directions.”
> > > > > > >     * Thank you for pointing this out. We have updated this statement in our paper to more clearly reflect how our findings are relevant for practitioners.
> > > > > > > *   “But I am not convinced of the current claim in main contribution #1 that "test accuracy of convolutional networks approaches that of fully connected networks as depth increases".”
> > > > > > >     * Thank you for pointing this out. We have updated main contribution #1 on page 2 to be more precise.  In particular, our main message is that test performance of CNNs gets worse with increasing depth and that in several cases, the performance of deep CNNs is actually worse than that of fully connected networks for image classification problems.  This is an important finding for practitioners since over-parameterized CNNs are generally expected to perform well for image classification problems.
> > > > > > > * ”Discussion in section 4 definitely implies convergence of solutions [...] I would prefer the wording around what exactly you're conjecturing and how exactly you corroborate your conjecture to be more precise.  [...] If, per your clarification, you do not claim that CNN solution converges to the MLP solution, then how else can the metrics of test accuracy and Frobenius norm be connected? In this case I don't see how Frobenius norm results give any intuition. But if you do not conjecture that they converge to the same solution, then how are Frobenius norm measurement related to accuracies?“
> > > > > > >     * Just to clarify further, our main goal is to understand why CNNs perform poorly at large depths (and not necessarily to claim that they converge to the same solution as an MLP in the nonlinear setting).  Analyzing the Frobenius norm of the resulting operator is relevant to understanding generalization in linear neural networks in cases where the solutions with lower Frobenius norm do not generalize.  In particular, we observe that deep linear CNNs produce solutions of lower norm with increasing depth and thus generalize worse.  We use the linear fully connected network in this case to demonstrate that the minimum Frobenius norm solution generalizes poorly.  We initially wrote our explanation in terms of solutions and not performance since we observed that deep linear CNNs learned solutions of decreasing norm and if this trend continued, deep linear CNNs would have to converge to the unique minimum norm solution.  However, we have clarified the text now to indicate just that deeper linear CNNs learn functions of decreasing Frobenius norm, which is expected to generalize worse.
> > > > > > > * “I would still prefer a similar experiment to be run on a small subset of CIFAR10 to be evaluated in a less artificial setting.”
> > > > > > >     * Thank you for the suggestion. We in fact did try this earlier, but found training deep linear CNNs on two classes of CIFAR10 to be computationally inefficient and difficult to optimize.  We are happy to start running these experiments, but due to the need to run several depths and random seeds, we are not sure if we can have these ready by the coming rebuttal deadline.
> > > > > > > * “Similarly to CNN vs MLP plots, in both Figures 7 and 19, I am not fully convinced the norm of the operator really converges to the minimum norm, instead of, for example, plateauing, which is arguably happening in Figure 7.c.”
> > > > > > >     * To clarify further, the point of this figure is to mainly demonstrate that deep linear CNNs learn solutions of decreasing Frobenius norm, which generalize worse for the given problem.  We have updated the main text to clarify this further.

---

> > > > > > > > ### Author Response · Authors · 2020-11-23
> > > > > > > > **Response Part 2/2**
> > > > > > > >
> > > > > > > > * “While I believe that the norm can change during training, I am not convinced that the resulting norm does not depend at all on the initial norm. I may be missing something here though, so please correct me if so.”
> > > > > > > >     * Thank you for clarifying your question further.  We definitely agree that the norm of the learned solution is very dependent on the initial norm.  However, our findings in the linear setting are surprising since training still leads to a solution of low norm.  As you mention, the norm can change significantly through training and an example is provided by single layer convolutional autoencoders.  Namely, if we have a shallow linear convolutional network (for example 2 or 3 layers on a large input), the resulting linear operator will necessarily be sparse.  Hence, regardless of how small we initialize our parameters, training will lead to solutions of high rank.  This is described more precisely in Section 4 of https://arxiv.org/abs/1810.10333.
> > > > > > > > * “However, note that Figure 4.a, GS4 shows the same trend for regular normally-initialized networks, and should probably be also discussed.”
> > > > > > > >     * Thank you for pointing this out.  We have now added a remark on this in our related work section.
> > > > > > > > * Regarding the NTK in our experiments
> > > > > > > >     * Thank you for sharing your code.  Using this, we were able to correct our NTK code and and have updated the paper to reflect these changes.
> > > > > > > > * “It appears that large depth behavior of the CNTK, and notably its generalization and implications for finite networks has been studied in https://arxiv.org/abs/1912.13053 (e.g. Figure 2), and I believe drawing connections to this work both in the context of this particular Figure and this paper in general would be very useful.”
> > > > > > > >     * Thank you for sharing this work.  We have included a reference to their work in our related work section.  This work actually claims that the CNTK is approximately equal to the NTK at large depths (page 8, section 6.4).  In our work, we instead focus on demonstrating that the test error for the CNTK first decreases monotonically and then increases monotonically.  In fact, in our experiments the test performance of the CNTK is actually worse than the NTK (even at 1000 depth).  Moreover, while https://arxiv.org/abs/1912.13053 claims that the NTK and CNTK are approximately equal at large depths, our figure 6 demonstrates this can manifest as a very large difference in test accuracy.

---

> > > > > > > > > ### Comment · AnonReviewer4 · 2020-11-24
> > > > > > > > > **Thank you for clarifications; last-minute questions**
> > > > > > > > >
> > > > > > > > > Thank you for such quick and elaborate responses and updates! Below some quick follow-up questions / concerns to further understand the CNN-MLP intuition.
> > > > > > > > >
> > > > > > > > > > Just to clarify further, our main goal is to understand why CNNs perform poorly at large depths (and not necessarily to claim that they converge to the same solution as an MLP in the nonlinear setting). Analyzing the Frobenius norm of the resulting operator is relevant to understanding generalization in linear neural networks in cases where the solutions with lower Frobenius norm do not generalize. In particular, we observe that deep linear CNNs produce solutions of lower norm with increasing depth and thus generalize worse. We use the linear fully connected network in this case to demonstrate that the minimum Frobenius norm solution generalizes poorly. We initially wrote our explanation in terms of solutions and not performance since we observed that deep linear CNNs learned solutions of decreasing norm and if this trend continued, deep linear CNNs would have to converge to the unique minimum norm solution. However, we have clarified the text now to indicate just that deeper linear CNNs learn functions of decreasing Frobenius norm, which is expected to generalize worse.
> > > > > > > > >
> > > > > > > > > I still don't really see how this gives intuition into why deep _nonlinear_ CNNs generalize poorly. Could you please complete the sentence "Deep nonlinear CNNs generalize poorly because --insert conjectured explanation here--" based on your findings? So far I cannot do it in my head.
> > > > > > > > >
> > > > > > > > > IIUC you've shown empirically that:
> > > > > > > > >   1. Deep nonlinear CNNs generalize poorly on image classification tasks.
> > > > > > > > >   2. Deep linear CNNs generalize poorly on a particular toy task, and also have a low Frobenius norm on this toy task where a low-norm solution is a poor solution.
> > > > > > > > >
> > > > > > > > > Firstly, per my comment below, I am still not convinced the trend to learn low-norm solutions is caused by depth and not something more trivial like using SAME zero padding.
> > > > > > > > >
> > > > > > > > > Secondly, I am not sure a low-norm solution will be a bad linear solution on an actual image task like a subset of CIFAR-10 (if anything, I might expect the opposite, which is why such an experiment could be particularly interesting). So I'm not even sure the claim "Deep linear CNNs generalize poorly because they learn solutions of small Frobenius norm" is true.
> > > > > > > > >
> > > > > > > > > And finally, even if we assume it to be true, I don't see how to make the logical step from the linear case to the nonlinear case. You can't use Frobenius norm as an explanation because the functions are not linear. You also can't claim convergence to the linear regression solution because we know this is not true, and as you've clarified, you're not claiming this anyway. So what is the underlying cause of deep nonlinear CNNs generalizing poorly in this case?
> > > > > > > > >
> > > > > > > > >
> > > > > > > > > > Thank you for clarifying your question further. We definitely agree that the norm of the learned solution is very dependent on the initial norm. However, our findings in the linear setting are surprising since training still leads to a solution of low norm. As you mention, the norm can change significantly through training and an example is provided by single layer convolutional autoencoders. Namely, if we have a shallow linear convolutional network (for example 2 or 3 layers on a large input), the resulting linear operator will necessarily be sparse. Hence, regardless of how small we initialize our parameters, training will lead to solutions of high rank. This is described more precisely in Section 4 of https://arxiv.org/abs/1810.10333.
> > > > > > > > >
> > > > > > > > > I still don't quite understand this. Firstly, this seems to be a result about autoencoding setting, and I don't see why it should transfer to regular classification tasks like CIFAR-10 or your toy classification task. Secondly, couldn't one have a matrix of full rank but arbitrarily small Frobenius norm by just multiplying it with a scalar? In this case, how does the rank of the matrix tell me anything about its norm?
> > > > > > > > >
> > > > > > > > > > Thank you for sharing this work. We have included a reference to their work in our related work section. This work actually claims that the CNTK is approximately equal to the NTK at large depths (page 8, section 6.4). In our work, we instead focus on demonstrating that the test error for the CNTK first decreases monotonically and then increases monotonically. In fact, in our experiments the test performance of the CNTK is actually worse than the NTK (even at 1000 depth). Moreover, while https://arxiv.org/abs/1912.13053 claims that the NTK and CNTK are approximately equal at large depths, our figure 6 demonstrates this can manifest as a very large difference in test accuracy.
> > > > > > > > >
> > > > > > > > > Just a quick comment here - per my reading of table 1 in their paper the claim about equivalence is for CNNs without pooling, while the CNN w/ pooling, which I assume is used in your experiments, would be the "CNN-P" in their paper, hence having different asymptotics (blue corrections in the table).

---

> > > > > > > > > > ### Author Response · Authors · 2020-11-24
> > > > > > > > > > **Thank you for the follow up! Part 1/3**
> > > > > > > > > >
> > > > > > > > > > Thank you again for the engaging and helpful discussion! We hope the following clarifies any confusion around our previous post.  Since we only have until the end of day today to respond, please let us know if there are any other points that you may find confusing, and we will do our best to answer as quickly as possible.
> > > > > > > > > >
> > > > > > > > > > * “I still don't really see how this gives intuition into why deep nonlinear CNNs generalize poorly. Could you please complete the sentence "Deep nonlinear CNNs generalize poorly because --insert conjectured explanation here--" based on your findings? So far I cannot do it in my head.”
> > > > > > > > > >     * We would complete the sentence as follows “Deep nonlinear CNNs generalize poorly because they learn low norm solutions in a larger function space.”  The underlying implication is that learning a low norm solution in a rich function space will not be ideal for certain problem settings such as image classification.  For example, the minimum norm solution may not be translation invariant, which is a desired property for image classification. This implication is reasonable since the community has consistently observed that deep nonlinear MLPs (which can express a large class of functions) perform poorly even in the infinite width setting (for example, this is also done the work you pointed out, https://arxiv.org/pdf/1912.13053.pdf).  We break down our intuition for the linear and nonlinear cases precisely below.  We don’t explicitly state this intuition in our work since we would first like to establish these results mathematically, but please let us know if this is a useful addition.
> > > > > > > > > >     * Our intuition comes exactly from comparing the solutions learned by linear CNNs and linear MLPs (and also is discussed briefly in the CNTK and NTK results in https://arxiv.org/abs/1912.13053).  In the linear setting, we know that a single layer linear fully connected network will learn the minimum Frobenius norm solution (when initialized at zero) and can express any linear operator given the input and output size. On the other hand, shallow linear CNNs cannot express any linear operator and in particular, cannot express the minimum Frobenius norm solution.  Nevertheless, increasing depth does increase expressivity (the set of operators which can be expressed as a depth $d$ convolutional network) in these networks and other networks with layer constraints (this is proven for Toeplitz matrices in https://www.stat.uchicago.edu/~lekheng/work/toeplitz.pdf).  Hence, with enough depth, these networks can express the minimum Frobenius norm solution. Indeed, our experiments indicate that these networks do learn solutions of increasingly low norm when increasing depth (and using an initialization starting at low norm).  Note in particular, that our linear CNNs do not involve any fully connected layers and thus the set of linear operators they can express is constrained, thus, it is not a priori guaranteed that these would learn low norm solutions.
> > > > > > > > > >     * For nonlinear infinite width networks, the RKHS for the NTK is known to be equivalent to the RKHS for the Laplace kernel (https://arxiv.org/pdf/2009.10683.pdf).  Our intuition is that without sufficient depth, the RKHS for the CNTK is more restricted and hence the minimum norm solution learned by kernel regression with shallow CNTKs will be significantly different from the minimum norm solution learned by the NTK.  The authors of https://arxiv.org/pdf/1912.13053.pdf (the latest related work you pointed us to) actually claim that as depth increases, the NTK and CNTK (without pooling) are approximately equal and we quote the relevant sentence on page 8):  “[The test performance of critically initialized CNNs degrades to that of MLPs] because (i) in the large width limit, the prediction of neural networks is characterized by the NTK and (ii) the NTKs of the two models are almost identical for large depth.”  Remarkably, our work demonstrates that a similar phenomenon seems to occur for finite width CNNs as well.  Moreover, we analyze how the test performance of the minimum norm solution in the RKHS for the CNTK (without pooling) changes with depth.

---

> > > > > > > > > > > ### Author Response · Authors · 2020-11-24
> > > > > > > > > > > **Part 2/3**
> > > > > > > > > > >
> > > > > > > > > > > * “Firstly, per my comment below, I am still not convinced the trend to learn low-norm solutions is caused by depth and not something more trivial like using SAME zero padding.”
> > > > > > > > > > >     * To clarify this further, using the SAME zero padding just ensures that the initialization has low norm.  This does not guarantee that the solution learned will also have a low norm. Without sufficient depth, the operator produced by a linear CNN is constrained (regardless of the setting of autoencoding or classification) since our network does not have any fully connected layers. Moreover, the minimum norm solution in the space of constrained matrices necessarily is larger than the minimum norm solution in the space of unconstrained matrices.  As an example, if we have a shallow (for example 1 layer CNN), the solution is necessarily going to be a doubly circulant sparse matrix.  This matrix necessarily has a higher norm than the minimum norm solution since parameters are shared along the rows (and thus, it is constrained).  This is what we observe happening in the linear case (even in our classification experiment).
> > > > > > > > > > >
> > > > > > > > > > > * “Secondly, I am not sure a low-norm solution will be a bad linear solution on an actual image task like a subset of CIFAR-10 (if anything, I might expect the opposite, which is why such an experiment could be particularly interesting). So I'm not even sure the claim ‘Deep linear CNNs generalize poorly because they learn solutions of small Frobenius norm’ is true.”
> > > > > > > > > > >     * Learning a minimum Frobenius norm solution would not capture properties of a solution that are useful for image classification (for example since the minimum norm solution would not be translation invariant).  This is what is demonstrated in our toy experiment: the minimum Frobenius norm solution is not translation invariant and thus generalizes poorly.  We do not claim that deep CNNs generalize poorly for all problems because of this, but rather that they generalize poorly for specific problems where the minimum Frobenius norm solution generalizes poorly.  Regarding the linearly separable subset of CIFAR-10, the construction of this set is important in understanding whether deep linear CNNs will perform well or not.  Namely, if we choose our subset such that it is separable by only high norm operators (i.e. translation invariance is important), then the minimum Frobenius norm solution would not generalize well (and so deep linear CNNs would not generalize well).  This is exactly what is demonstrated in our toy experiment: the data is separable with respect to a high norm operator and not the minimum norm operator (due to the translation invariance requirement).  In general, as translation invariance is important for image classification we would not expect low norm solutions to work for these problems either.
> > > > > > > > > > >
> > > > > > > > > > > * “And finally, even if we assume it to be true, I don't see how to make the logical step from the linear case to the nonlinear case. You can't use Frobenius norm as an explanation because the functions are not linear. You also can't claim convergence to the linear regression solution because we know this is not true, and as you've clarified, you're not claiming this anyway. So what is the underlying cause of deep nonlinear CNNs generalizing poorly in this case?”
> > > > > > > > > > >     * Regarding the nonlinear case, the NTK and CNTK provide an appropriate analog for why deep nonlinear CNNs generalize poorly.  Namely, the CNTK for deep CNNs is claimed to be approximately equal to the NTK, which yields a minimum norm solution in a rich RKHS and thus generalizes poorly. However, for the shallower CNTK we learn a minimum norm solution in a more constrained RKHS, which we observe generalizes better. This is analogous to how deep linear CNNs yield a low norm solution in a much larger class of linear operators than shallow CNNs and thus generalize poorly in cases where the minimum Frobenius norm solution generalizes poorly.
> > > > > > > > > > >
> > > > > > > > > > > * “I still don't quite understand this. Firstly, this seems to be a result about autoencoding setting, and I don't see why it should transfer to regular classification tasks like CIFAR-10 or your toy classification task. Secondly, couldn't one have a matrix of full rank but arbitrarily small Frobenius norm by just multiplying it with a scalar? In this case, how does the rank of the matrix tell me anything about its norm?”
> > > > > > > > > > >     * We are sorry for the confusion and hope the following helps to clarify.  The example of high rank was specifically for autoencoding where the “high rank” solution is the identity and thus has a higher norm.  In general, the solution learned by a shallow CNN will be constrained (see for example, Appendix E of https://arxiv.org/abs/1810.10333) and thus will necessarily have a higher norm than the minimum norm solution.

---

> > > > > > > > > > > > ### Author Response · Authors · 2020-11-24
> > > > > > > > > > > > **Part 3/3**
> > > > > > > > > > > >
> > > > > > > > > > > > * “...while the CNN w/ pooling, which I assume is used in your experiments…”
> > > > > > > > > > > >     * We actually use the CNTK implementation from https://arxiv.org/pdf/1904.11955.pdf without global average pooling (column “CNTK-V” in Table 1), which should correspond to “CNN-F” in https://arxiv.org/abs/1912.13053.

---

> > > > > > > > > > > > > ### Comment · AnonReviewer4 · 2020-11-24
> > > > > > > > > > > > > **Thank you! Last-minute, perhaps less-organized thoughts, part 1/3**
> > > > > > > > > > > > >
> > > > > > > > > > > > > Thank you for your quick replies! Posting some further follow-ups, they may be not fully thought-out / organized, but I'd rather get them out quickly.
> > > > > > > > > > > > >
> > > > > > > > > > > > > Overall, I now understand your intuition as
> > > > > > > > > > > > > 1. Conjecture/assumption: deep linear/nonlinear CNNs learn solutions of minimal norm in their search space, for some definition of norm. _Concern: then why does the norm saturate in Figure 7.c? What is the norm definition for nonlinear networks?_
> > > > > > > > > > > > > 2. As the search space is expanded via increasing depth, it begins to include poor solutions. If these poor solutions are of minimal norm in that space, deep CNNw will learn them, and not generalize. _Concern: why do we expect poor solutions of deep networks to have low norm on image tasks (I believe the argument about translation invariance is incorrect, see detailed comments below)?_
> > > > > > > > > > > > > 3. Therefore, shallow is better. _Concern: then why aren't best networks not always the most shallow among those with perfect training accuracy?_
> > > > > > > > > > > > >
> > > > > > > > > > > > > Please see some detailed responses below (potentially with some repetition, apologies for that).
> > > > > > > > > > > > >
> > > > > > > > > > > > > > We would complete the sentence as follows “Deep nonlinear CNNs generalize poorly because they learn low norm solutions in a larger function space.” The underlying implication is that learning a low norm solution in a rich function space will not be ideal for certain problem settings such as image classification.
> > > > > > > > > > > > >
> > > > > > > > > > > > > What norm do you have in mind for nonlinear functions? I am concerned that this explanation goes against the more common observation of low-norm networks (for some particular definitions of "norm") generalizing _better_, e.g. https://arxiv.org/pdf/1706.08947.pdf, Figure 3, center pane, or https://arxiv.org/pdf/1802.08760.pdf Figure 5, or just the common practice of using weight decay when training neural networks, which I imagine drives down most reasonable definitions of norms.
> > > > > > > > > > > > >
> > > > > > > > > > > > > Further, if you do have a certain notion of norm for this claim, how do you explain that CNNs that generalize best are not always the most shallow (as we've discussed above)?
> > > > > > > > > > > > >
> > > > > > > > > > > > >
> > > > > > > > > > > > > > For example, the minimum norm solution may not be translation invariant, which is a desired property for image classification.
> > > > > > > > > > > > >
> > > > > > > > > > > > > This depends on the search space. For example in your CIFAR-10/Imagenet32 experiments I assume you use global average pooling at the top. In this case all solutions will be translation-invariant, regardless of their norm, and regardless of the definition of the norm.
> > > > > > > > > > > > >
> > > > > > > > > > > > >
> > > > > > > > > > > > > > This implication is reasonable since the community has consistently observed that deep nonlinear MLPs (which can express a large class of functions) perform poorly even in the infinite width setting (for example, this is also done the work you pointed out, https://arxiv.org/pdf/1912.13053.pdf). We break down our intuition for the linear and nonlinear cases precisely below. We don’t explicitly state this intuition in our work since we would first like to establish these results mathematically, but please let us know if this is a useful addition.
> > > > > > > > > > > > >
> > > > > > > > > > > > > I believe MLPs generalize poorly regardless of depth (even in your paper), and IIUC the argument made in https://arxiv.org/pdf/1912.13053.pdf is about numerics / trainability and not expressivity - those kernels of any depth can express any function possible, so they are all equally expressive (unless you introduce a different, more specific notion of expressivity). This is just to point out that I don't think high expressivity has been shown to be problematic for networks in prior works, and could be an interesting angle to position this work rather than the current norm-based intuition, which I still find somewhat fragile.
> > > > > > > > > > > > >
> > > > > > > > > > > > >
> > > > > > > > > > > > >
> > > > > > > > > > > > >
> > > > > > > > > > > > > > Our intuition comes exactly from comparing the solutions learned by linear CNNs and linear MLPs (and also is discussed briefly in the CNTK and NTK results in https://arxiv.org/abs/1912.13053). In the linear setting, we know that a single layer linear fully connected network will learn the minimum Frobenius norm solution (when initialized at zero) and can express any linear operator given the input and output size. On the other hand, shallow linear CNNs cannot express any linear operator and in particular, cannot express the minimum Frobenius norm solution. Nevertheless, increasing depth does increase expressivity (the set of operators which can be expressed as a depth  convolutional network) in these networks and other networks with layer constraints (this is proven for Toeplitz matrices in https://www.stat.uchicago.edu/~lekheng/work/toeplitz.pdf).
> > > > > > > > > > > > >
> > > > > > > > > > > > > Thank you. I see your intuition that shallow CNNs may be restricted to a certain class of functions that can exclude poor solutions. But again, I don't really see why the notion of norm is important here, rather than just saying that shallow CNNs are constrained to exclude certain poor solutions in this particular case. As I comment below, I am not fully convinced deep CNNs necessarily learn the min-norm solution.

---

> > > > > > > > > > > > > > ### Comment · AnonReviewer4 · 2020-11-24
> > > > > > > > > > > > > > **Part 2/3**
> > > > > > > > > > > > > >
> > > > > > > > > > > > > > > Hence, with enough depth, these networks can express the minimum Frobenius norm solution. Indeed, our experiments indicate that these networks do learn solutions of increasingly low norm when increasing depth (and using an initialization starting at low norm).
> > > > > > > > > > > > > >
> > > > > > > > > > > > > > If this were true, shouldn't the norm of the operator in Figure 7.c keep decreasing beyond depth 20, and ultimately reach the min-norm solution?
> > > > > > > > > > > > > >
> > > > > > > > > > > > > > > Note in particular, that our linear CNNs do not involve any fully connected layers and thus the set of linear operators they can express is constrained, thus, it is not a priori guaranteed that these would learn low norm solutions.
> > > > > > > > > > > > > >
> > > > > > > > > > > > > > Could you please clarify how exactly you get the output? I was assuming a fully connected, and / or global average pooling. Do you do stride > 1 convolutions in top few layers, or use valid padding in top few layers, to reduce the output size to 1?
> > > > > > > > > > > > > >
> > > > > > > > > > > > > >
> > > > > > > > > > > > > >
> > > > > > > > > > > > > > > For nonlinear infinite width networks, the RKHS for the NTK is known to be equivalent to the RKHS for the Laplace kernel (https://arxiv.org/pdf/2009.10683.pdf). Our intuition is that without sufficient depth, the RKHS for the CNTK is more restricted and hence the minimum norm solution learned by kernel regression with shallow CNTKs will be significantly different from the minimum norm solution learned by the NTK.
> > > > > > > > > > > > > >
> > > > > > > > > > > > > > Unfortunately I don't have time to get to know this work well before the deadline. But so far I don't see how the norm is relevant here (in both cases you compare minimum-norms solutions of different RKHS), and I'm not sure what exactly "restricted" means (AFAIK any NTK kernel can express any finite dataset), and how this would be related to the notion of norm.
> > > > > > > > > > > > > >
> > > > > > > > > > > > > >
> > > > > > > > > > > > > > > The authors of https://arxiv.org/pdf/1912.13053.pdf (the latest related work you pointed us to) actually claim that as depth increases, the NTK and CNTK (without pooling) are approximately equal and we quote the relevant sentence on page 8): “[The test performance of critically initialized CNNs degrades to that of MLPs] because (i) in the large width limit, the prediction of neural networks is characterized by the NTK and (ii) the NTKs of the two models are almost identical for large depth.” Remarkably, our work demonstrates that a similar phenomenon seems to occur for finite width CNNs as well. Moreover, we analyze how the test performance of the minimum norm solution in the RKHS for the CNTK (without pooling) changes with depth.
> > > > > > > > > > > > > >
> > > > > > > > > > > > > > Please note that per our discussion above, I still believe your work shows degradation of performance with depth, but _not convergence to MLPs_ (neither in terms of solutions, nor accuracy; also, you have pooling in most settings), while this works talks about precisely convergence, so I would suspect the reason behind your results is different, which I find interesting (but at the same time, I still don't think it's explained via the discussion around norms).
> > > > > > > > > > > > > >
> > > > > > > > > > > > > >
> > > > > > > > > > > > > > > To clarify this further, using the SAME zero padding just ensures that the initialization has low norm. This does not guarantee that the solution learned will also have a low norm.
> > > > > > > > > > > > > >
> > > > > > > > > > > > > > Sure, however I was pointing out that it is likely (intuitively, no rigorous argument here, but I also believe there's no rigorous argument in the other direction) to bias the solution to a low norm solution, and hence it would be best to control for this effect.
> > > > > > > > > > > > > >
> > > > > > > > > > > > > > > Without sufficient depth, the operator produced by a linear CNN is constrained (regardless of the setting of autoencoding or classification) since our network does not have any fully connected layers. Moreover, the minimum norm solution in the space of constrained matrices necessarily is larger than the minimum norm solution in the space of unconstrained matrices. As an example, if we have a shallow (for example 1 layer CNN), the solution is necessarily going to be a doubly circulant sparse matrix. This matrix necessarily has a higher norm than the minimum norm solution since parameters are shared along the rows (and thus, it is constrained). This is what we observe happening in the linear case (even in our classification experiment).
> > > > > > > > > > > > > >
> > > > > > > > > > > > > > Thank you for your explanation. I agree that min-norm in a constrained space is higher norm than min-norm in a less constrained space. But for instance Figure 7.c arguably shows that CNNs don't really learn the min-norm solution (or at least this is a rather speculative claim), otherwise I'd expect the norm of operator to continue decreasing past depth 20. For this reason I suspect the initial norm matters, and would prefer if it was decoupled from the findings here.

---

> > > > > > > > > > > > > > > ### Comment · AnonReviewer4 · 2020-11-24
> > > > > > > > > > > > > > > **Part 3/3**
> > > > > > > > > > > > > > >
> > > > > > > > > > > > > > > > Learning a minimum Frobenius norm solution would not capture properties of a solution that are useful for image classification (for example since the minimum norm solution would not be translation invariant). This is what is demonstrated in our toy experiment: the minimum Frobenius norm solution is not translation invariant and thus generalizes poorly. We do not claim that deep CNNs generalize poorly for all problems because of this, but rather that they generalize poorly for specific problems where the minimum Frobenius norm solution generalizes poorly. Regarding the linearly separable subset of CIFAR-10, the construction of this set is important in understanding whether deep linear CNNs will perform well or not. Namely, if we choose our subset such that it is separable by only high norm operators (i.e. translation invariance is important), then the minimum Frobenius norm solution would not generalize well (and so deep linear CNNs would not generalize well). This is exactly what is demonstrated in our toy experiment: the data is separable with respect to a high norm operator and not the minimum norm operator (due to the translation invariance requirement). In general, as translation invariance is important for image classification we would not expect low norm solutions to work for these problems either.
> > > > > > > > > > > > > > >
> > > > > > > > > > > > > > > Thank you for further clarifications. Unfortunately I still believe that this is the kind of result that would be nice to have evaluated experimentally, since per my comment above, I am not sure deep linear CNNs necessarily learn the min-norm solutions. Further, please note that I believe if you have only convolutions in this architecture (no fully connected layers), I think the operator has to be translation invariant by definition, regardless of depth, before or after training, and the only aspect breaking the precise translation invariance would be exactly the edge effects due to SAME zero padding. If you were to make each layer circularly padded, I believe the network would become precisely translation invariant, and this would be independent of depth, norm, training etc. Similarly, as I mentioned above, in image classification experiments pooling is used, which again makes the network translation invariant, and invariance breaking would be due to edge effects due to SAME padding, but not due to having the solution be low norm. This is another reason it would be nice to disentangle this edge padding effect from the discussion about the norm.
> > > > > > > > > > > > > > >
> > > > > > > > > > > > > > > > Regarding the nonlinear case, the NTK and CNTK provide an appropriate analog for why deep nonlinear CNNs generalize poorly. Namely, the CNTK for deep CNNs is claimed to be approximately equal to the NTK, which yields a minimum norm solution in a rich RKHS and thus generalizes poorly. However, for the shallower CNTK we learn a minimum norm solution in a more constrained RKHS, which we observe generalizes better. This is analogous to how deep linear CNNs yield a low norm solution in a much larger class of linear operators than shallow CNNs and thus generalize poorly in cases where the minimum Frobenius norm solution generalizes poorly.
> > > > > > > > > > > > > > >
> > > > > > > > > > > > > > > Thank you, I can see the intuition about finite linear networks, where indeed, shallow networks are restricted to a certain class of solutions, that may exclude poor solutions like the min-norm here. But I don't see in which sense is the RKHS of a deep CNTK richer than that of a shallow CNTK, since both can express any function?
> > > > > > > > > > > > > > >
> > > > > > > > > > > > > > >
> > > > > > > > > > > > > > > > We are sorry for the confusion and hope the following helps to clarify. The example of high rank was specifically for autoencoding where the “high rank” solution is the identity and thus has a higher norm. In general, the solution learned by a shallow CNN will be constrained (see for example, Appendix E of https://arxiv.org/abs/1810.10333) and thus will necessarily have a higher norm than the minimum norm solution.
> > > > > > > > > > > > > > >
> > > > > > > > > > > > > > > Understood, thanks.
> > > > > > > > > > > > > > >
> > > > > > > > > > > > > > >
> > > > > > > > > > > > > > > > We actually use the CNTK implementation from https://arxiv.org/pdf/1904.11955.pdf without global average pooling (column “CNTK-V” in Table 1), which should correspond to “CNN-F” in https://arxiv.org/abs/1912.13053.
> > > > > > > > > > > > > > >
> > > > > > > > > > > > > > > Understood, thanks.

---

> > > > > > > > > > > > > > > > ### Author Response · Authors · 2020-11-25
> > > > > > > > > > > > > > > > **Thanks for the discussion**
> > > > > > > > > > > > > > > >
> > > > > > > > > > > > > > > > Thank you again for the discussion!  While very interesting, some of the discussion is now drifting away from the main message of our work.  Our current discussion seems to be centering around the intuition for how Section 4 supports the rest of our work. However, we would like to emphasize that the main contributions of our work are the experiments identifying a monotonic decrease and then increase in test error in CNNs of increasing depth, which to our knowledge have not been the main focus of any prior works.  The intuition regarding minimum norm solutions is based on results in several previous works, but as there are no concrete mathematical proofs of any of these claims, it is difficult to say anything with certainty.  We have aimed to keep our paper’s main points precise and not make any concrete claims that are not backed up by either our experiments or previous theory, but please do let us know if you feel this is not the case.  We thank you again for the discussion, which has been very valuable.  If you feel that we have better addressed your original concerns and that our paper is stronger in light of the new experiments, we hope that you would consider increasing your score.
> > > > > > > > > > > > > > > >
> > > > > > > > > > > > > > > > Given the time constraints, we focus our response on your 3 main concerns below:
> > > > > > > > > > > > > > > >
> > > > > > > > > > > > > > > > * “1. Conjecture/assumption: deep linear/nonlinear CNNs learn solutions of minimal norm in their search space, for some definition of norm. _Concern: then why does the norm saturate in Figure 7.c?  What is the norm definition for nonlinear networks?_  [...] shouldn't the norm of the operator in Figure 7.c keep decreasing beyond depth 20, and ultimately reach the min-norm solution?”
> > > > > > > > > > > > > > > >     * To clarify, the norm does not saturate in Figure 7c.  The figure just appeared that way due to our random samples.  We have updated this figure to further demonstrate this by adding even more depth.  Note that since we cannot initialize all layers exactly at zero, there will always be some small term corresponding to random initialization that cannot be removed (which is consistent with the case of random initialization in linear regression).
> > > > > > > > > > > > > > > >      * For nonlinear networks, we had in mind the norm in the corresponding RKHS.
> > > > > > > > > > > > > > > > * “2. As the search space is expanded via increasing depth, it begins to include poor solutions. If these poor solutions are of minimal norm in that space, deep CNNs will learn them, and not generalize. _Concern: why do we expect poor solutions of deep networks to have low norm on image tasks (I believe the argument about translation invariance is incorrect, see detailed comments below)?_”
> > > > > > > > > > > > > > > >     * We don’t expect all poor solutions to have low norm; however, since the fully connected solution is poor, and has low norm, our intuition is that the low norm solutions are not optimal as they will be close to the fully connected solution.  This is in line with the intuition given on page 8, section 6.4 of https://arxiv.org/pdf/1912.13053.pdf as to why the CNTK for deep CNNs performs similarly to that of deep NTK.
> > > > > > > > > > > > > > > > * “3. Therefore, shallow is better. _Concern: then why aren't best networks not always the most shallow among those with perfect training accuracy?_”
> > > > > > > > > > > > > > > >     * We would like to emphasize again that identifying the critical depth threshold of best performance is not a main message of our work, but we regard it as an important future work.  Our main goal is to just demonstrate the phenomenon of single (and not double) descent in the test error with increasing depth regardless of whether we are in the CNTK or in the finite width setting. We are not claiming that shallow is always better, but rather that increasing depth significantly will lead to worsening performance unlike double descent.  Our intuition from the linear setting is that the minimum norm solution can perform poorly, not necessarily that a smaller norm always manifests as having worse performance.
> > > > > > > > > > > > > > > >
> > > > > > > > > > > > > > > > Some minor clarifications:
> > > > > > > > > > > > > > > >
> > > > > > > > > > > > > > > > * “Could you please clarify how exactly you get the output?”
> > > > > > > > > > > > > > > >     * Our final convolutional layer has 1 output channel, and then we average all the values in the channel to get our prediction.
> > > > > > > > > > > > > > > >
> > > > > > > > > > > > > > > > * “I don't see in which sense is the RKHS of a deep CNTK richer than that of a shallow CNTK”
> > > > > > > > > > > > > > > >     * It’s not true that an RKHS can contain any function; the introduction of https://arxiv.org/pdf/2009.10683.pdf discusses how different kernels lead to RKHS’s of different sizes.

---

### Official Review · AnonReviewer3 · 2020-10-28
**Comments**

**Rating:** 5
**Confidence:** 3

**Review:**

*Summary:

This paper mainly answers a fundamental question: what is the role of depth in convolutional networks? Specifically, the authors present an empirical analysis of the impact of the depth on the generalization in CNNs. Experiments on CIFAR10 and ImageNet32 demonstrate that the test performance beyond a critical depth. My detailed comments are as follows.

*Positive points:

1. This paper is significant to understand deep neural networks and helps to develop new deep learning algorithm.

2. This paper provides many empirical studies to analyze the effect of increasing depth on test performance.

*Negative points:

1. The importance and novelty of the research should be emphasized. Recently, there are some works [1][2][3] study the role of depth in DNN. What is the difference from these works?

[1] Do Deep Convolutional Nets Really Need to Be Deep and Convolutional? ICLR 2017
[2] Understanding intermediate layers using linear classifier probes. arXiv, 2016.
[3] Towards Interpreting Deep Neural Networks via Understanding Layer Behaviors. 2020

2. This paper analyzes the linear neural networks and demonstrates that increasing depth leads to poor generalization. However, existing works apply non-linear neural networks in real-world case. It would be better to provide analysis on non-linear neural networks.

3. The authors suggest that practitioners should decrease depth in these settings to obtain better test performance. However, ResNet-101 has better test performance than ResNet-18 in practice. Could you please give more explanations?

---

> ### Author Response · Authors · 2020-11-18
> **Response to Reviewer 3**
>
> We thank the reviewer for their feedback. We address the stated negative points below:
> * “Recently, there are some works [1][2][3] study the role of depth in DNN. What is the difference from these works?”
>     * While we already discuss Urban et al. (2017) in our related works section, we clarify further below.  As mentioned in our related work section, our work differs from the analysis conducted in Urban et al (2017). That paper studies the role of depth in student-teacher CNNs, where a “shallow” CNN (defined as having fewer than 5 layers) is trained to fit the logits of an ensemble of deep CNNs (which only has 8 convolutional layers and is far smaller than both currently used CNNs and the models considered in our work). Our work, on the other hand analyzes the performance of models with increasing depth that are trained from scratch. Therefore the student-teacher analysis of Urban et al (2017) doesn’t apply here. We have updated the related work section to make this distinction clearer.
>     *[2] and [3] do not appear to be relevant to the main topic of this current work, as both are related to neural network interpretability.  In particular, both [2] and [3] attempt to understand the representations learned through intermediate layers of a deep network, while we study the test performance of networks of varying depth trained from scratch.
> * “This paper analyzes the linear neural networks and demonstrates that increasing depth leads to poor generalization. However, existing works apply non-linear neural networks in real-world case. It would be better to provide analysis on non-linear neural networks.”
>     * The bulk of our experimental analysis (provided in section 3) is already conducted on non-linear neural networks: the Fully-conv net, ResNets, and the CNTK. We demonstrate for these examples that increasing depth leads to worsening generalization.
> * “The authors suggest that practitioners should decrease depth in these settings to obtain better test performance. However, ResNet-101 has better test performance than ResNet-18 in practice. Could you please give more explanations?”
>     * The claim that ResNet-101 has better test-performance than ResNet-18 depends on a number of factors such as network width and the classification problem at hand.  For CIFAR10, as shown in plot 1(b), the width 64 ResNet-18 actually has the best test performance of all models considered. However, for models with smaller widths, the optimal depth model is somewhere between 18 and 101, and thus it is certainly possible for ResNet-101 to have better test performance. Our claim is that there exists a critical depth and, beyond that critical depth, test performance begins to degrade.
>     * If the reviewer is referring to ResNet performance on ImageNet, we note that the reported performance of ResNet-101 does not actually correspond to a training accuracy of 100%.  Hence, in accordance with our results, it is plausible that increasing depth for ResNet on full ImageNet could lead to better generalization.
>
> Lastly, we would like to mention that we have added in a number of new experiments:
>
> 1.  We have added a subsection of experiments (Section 3.3, Appendix D.6) in the infinite-width regime, using the Convolutional Neural Tangent Kernel (CNTK). The CNTK is deterministic (eliminating the need for random seed sampling), and perfectly interpolates the data at all depths. We again observe that the test loss is monotonically increasing beyond a critical depth. Our experiments additionally show that for large depths the performance of the CNTK is comparable to that of the NTK.
> 2.  We have run our main ResNet experiment at depths up to 100 (Figures 5, 13 and 14), which makes clearer the trend that loss is increasing beyond a critical depth.
> 3. We ran additional experiments for the toy dataset in Section 4 and Appendix D.7.

---

### Official Review · AnonReviewer1 · 2020-10-29
**Review of the manuscript Do Deeper Convolutional Networks Perform Better?**

**Rating:** 6
**Confidence:** 5

**Review:**

*********
Summary Of The Manuscript:
*********
This manuscript focuses on the problem of the impact of depth in Convolutional Neural Networks (CNNs) for better generalization. To investigate the issue, the authors did an empirical analysis of the problem statement through various strategies (i.e. Deep CNNs - increasing depth vs. Fully Convolutional Networks, etc.) and provided in-depth findings via experiments. Together with experiments and evaluation on standard benchmarks such as CIFAR-10 and ImageNet32, the authors lead to the conclusion of their study that the testing performance will be poor if the increase in depth is higher.

*********
Strength Of The Manuscript:
*********
Clarity:
++ The paper reads very well and provides a very good description of related work and background, motivating the problem. Even outside of the contribution of this paper, I would recommend this paper to people getting started with CNNs as it provides a thorough description of the part of the pipelines it deals with. Also, all the empirical analyses have been described thoroughly and the various settings for the training and testing are performed in such a way to give better insights to novice readers.

Novelty:
++ In terms of novelty, the stand-alone contribution of the manuscript is that through various strategies authors tried to give in-depth insights on the behavior of different linear and non-linear as well as deep learning models via increasing the depth for classification task on different benchmark datasets.

Experiments:
++ There have been experiments performed across datasets with variety in terms of different models and their architectures. Additionally, the analyses provided in the manuscript is fairly consistent. Supplementary material also backs the analysis by providing the visualizations of training and testing errors. Besides, analysis/ablation studies are done on the models, by changing the widths, the effect of downsampling, and changing the kernel width, and after through all the settings finally check the performance difference.

*********
Weakness Of The Manuscript:
*********
Overall, apart from the contribution of the paper I have some concerns regarding the paper which are listed below.

-- I believe that the authors did a tremendous job to reach the conclusion that the increase in depth is crucial for certain tasks and might lead to poor results in different situations if the depth is increased beyond the threshold. However, to back up this conclusion more effectively I believe that if the authors have performed more analysis by introducing a data-augmentation strategy,student-teacher training strategy, introducing learning rate variance and so more training strategies this manuscript will be a really a good point of start to a beginner. I encourage the authors to refer to this manuscript by Urban et al. [1].
-- The fact that has been mentioned corresponding to increase in depth leads to worsening the result have already been a point of view for much of the deep learning practitioners as He et al. [2] have already provided a really good analysis in the manuscript of ResNet, in contrast, the majority of the work in the current manuscript is already have been either published or have been known to the community.

*********
Justification Of The Manuscript Review:
*********
-- In the reviewer's opinion, in its current form, the paper provides in-depth analysis for increasing /decreasing the depth of CNNs however there are certain points mentioned in the weakness section have been mentioned which needs some clarification from the authors during the rebuttal phase, if answered thoroughly, the reviewer believes that the manuscript will be a really good point of start for novice deep learning readers/practitioners. Therefore the current rating of the paper will be 6 in reviewers' opinion as it is above the acceptance threshold marginally.

References:
[1] Urban, Gregor, et al. "Do deep convolutional nets really need to be deep and convolutional?." In ICLR (2017).
[2] He, Kaiming, et al. "Deep residual learning for image recognition." Proceedings of the IEEE conference on computer vision and pattern recognition. 2016.

---

> ### Author Response · Authors · 2020-11-18
> **Response to Reviewer 1**
>
> We thank the reviewer for their feedback and their positive comments. We would like to address the stated weaknesses and discuss new experiments that we added to our paper:
> * “However, to back up this conclusion more effectively I believe that if the authors have performed more analysis by introducing a data-augmentation strategy,student-teacher training strategy, introducing learning rate variance and so more training strategies this manuscript will be a really a good point of start to a beginner.”
>     * Our experiments with ResNets in Section 3.2 follow the methodology of Yang et al (2020), and do use the data augmentation strategy of random crops and random horizontal flips. We have updated the paper to make this explicit.
>     * As mentioned in our related work section, our work differs from the analysis conducted in Urban et al (2017). That paper studies the role of depth in student-teacher CNNs, where a “shallow” CNN (defined as having fewer than 5 layers) is trained to fit the logits of an ensemble of deep CNNs (which only has 8 convolutional layers and is far smaller than both currently used CNNs and the models considered in our work). Our work, on the other hand analyzes the performance of models with increasing depth that are trained from scratch. Therefore the student-teacher analysis of Urban et al (2017) doesn’t apply here. We have updated the related work section to make this distinction clearer.
>     * We have added a subsection of experiments (Section 3.3, Appendix D.6) in the infinite-width regime, using the Convolutional Neural Tangent Kernel (CNTK). The CNTK is deterministic, and perfectly interpolates the data at all depths. We again observe that the test loss is monotonically increasing beyond a critical depth. Our experiments additionally show that for large depths the performance of the CNTK is comparable to that of the NTK.
>     * Could you clarify what you mean by “introducing learning rate variance”? As mentioned in Appendix C, our models are trained either using Adam, or SGD with momentum with a learning rate schedule.
> * “The fact that has been mentioned corresponding to increase in depth leads to worsening the result have already been a point of view for much of the deep learning practitioners as He et al. [2] have already provided a really good analysis in the manuscript of ResNet, in contrast, the majority of the work in the current manuscript is already have been either published or have been known to the community.”
>     * We would like to clarify that our experiments are in the over-parameterized regime and are thus in stark contrast with those from He et al. 2015, which generally do not interpolate (achieve 100% training accuracy) on the training set.  The experiments in He et al. (2015) address the fact that increasing depth leads to training difficulty.  For example, Figure 1 from their paper presents an example of a 56 depth network having higher training error than a 20 depth network.  However, none of our networks have any issues with training and are able to perfectly interpolate the data. He et al. (2015) in fact has only 1 example demonstrating that deeper interpolating models perform worse (ResNet 1202 on CIFAR10).  Our work presents a systematic analysis and demonstrates that this phenomenon occurs across a number of convolutional architectures. Additionally, our result is surprising in light of the double descent results regarding network width.

---

### Official Review · AnonReviewer2 · 2020-11-06
**Empirical work that studies an interesting question -- could incorporate a bit more content**

**Rating:** 6
**Confidence:** 4

**Review:**

Summary of paper: The authors empirically study trends in neural network performance for models with fixed width but increasing depth. (Previous work has investigated how increasing "model complexity" from neural network width affects test loss/accuracy, with "double descent" behavior -- the trends observed here are very different from double descent.)

The authors consider ResNets and fully-convolutional networks (which contain only convolutional layers with a final pooling operation) on CIFAR-10 and subsets of ImageNet32. One primary finding is that the test accuracy of convolutional networks approaches that of fully-connected networks as depth increases (Fig. 2). These experiments include sweeping model complexity past the "interpolation threshold" (where 100% train accuracy is achievable), by analogy with experiments that have observed double descent behavior when model complexity originates from width.

The authors further study linear neural networks (with convolutional or Toeplitz constraints against fully-connected layers) where they can analyze properties of the learned solutions. In one experiment, they consider a toy problem (Sect. 4.1) of classifying the color of a single pixel in an image. This problem is chosen because the minimum Frobenius norm solution (which is learned by the fully-connected network) does not generalize, and the authors show that the norm of the solution learned by the linear convolutional network approaches this value with depth. A similar result is shown for linear autoencoders (Fig. 7).

A primary implication of the authors' results is that increasing depth past the interpolation threshold may be detrimental to performance.

Quality and Clarity: The quality of the work (experiments and framing) seems good, as far as I can gather, and the paper is clearly written and straightforward to read.

Originality: The primary thrust of the paper (studying the trend in loss/accuracy when model complexity originates from depth) is in close analogy to prior works that have investigated network width and not especially novel. However, I did find the comparison between learned solutions of linear convolutional vs fully-connected networks to be interesting and different from prior works I'm aware of. The class-dependence of the critical depth was also an interesting finding.

Significance: I think the work brings an interesting basic question to light -- to what extent is behavior like double descent (or the usual bias-variance tradeoff) specific to the way in which model complexity is increased in neural networks, and it provides empirical support that the trends are very different when overparameterization arises from depth.

Other comments: The observation that the performance of deep convolutional networks approaches that of a fully-connected networks has appeared in one prior work that I'm aware of: https://arxiv.org/abs/1806.05393 published in ICML 2018. Fig. 3 in that work shows the test performance of ultra-deep CNNs (with an initialization scheme chosen specifically to help train ultra-deep networks) on CIFAR-10 collapses to that of fully-connected networks; Sect. 2.1.6. discusses how the behavior of signal propagation (the model prior) in random convolutional networks as they become deep approaches that of fully-connected architectures.
Since I was familiar with this work, I am not especially surprised by the findings in this paper and hence the reason for my somewhat lower score and suggestion for incorporating a bit more (richer) content, either empirically or theoretically. I do think the results are interesting to publish in some form but am not sure the paper as it stands merits a conference rather than e.g. a workshop publication. I am open to raising my score however.

I had several questions about the empirical results:
--Some of the trends seem to fluctuate a bit rather than behave smoothly after an inflection point (e.g. Fig. 1(b)). Is there a reason for this?
--Could the authors elaborate on why a lower Frobenius norm (in the linear networks) is a bad solution? In general I might have naively thought that low norm solutions were good for generalization.
--In Fig. 2(e) the test accuracy of the fully-convolutional network actually dips below that of the fully-connected model. This seems a bit unexpected -- is there an understanding of why this occurs?

Finally, Fig. 14 in the supplementary seems to be mislabeled (y-axis reads "Test MSE", caption refers to train loss).

---

> ### Author Response · Authors · 2020-11-18
> **Response to Reviewer 2**
>
> We thank the reviewer for their feedback and their positive comments. Below, we address your concerns and describe additional experiments that we have included:
> * “The observation that the performance of deep convolutional networks approaches that of a fully-connected networks has appeared in one prior work that I'm aware of: https://arxiv.org/abs/1806.05393 published in ICML 2018.”
>     * We thank the reviewer for pointing out the related work, and we have updated our related work section in the paper to discuss Xiao et al. (2018).  Figure 3 of Xiao et al. (2018) demonstrates that  deeper models perform worse, for the specific initialization scheme considered in their paper.  The paper hypothesizes that the decrease in test performance is due to “attenuation of spatially non-uniform modes,” which would be an artifact of the initialization scheme chosen. Our work shows that the decrease in test performance is a universal phenomenon in models used in practice.
>     * While Xiao et al (2018) argues that the signal propagation for infinitely-wide, random convolutional networks approaches that of fully-connected networks, this does not give insight on the comparison of these two models after training.  We have added additional experiments in the Convolutional Neural Tangent Kernel setting (Section 3.3) to show that this phenomenon does indeed occur in the infinite-width regime after training.
>
> To address the additional questions:
> * “Some of the trends seem to fluctuate a bit rather than behave smoothly after an inflection point (e.g. Fig. 1(b)). Is there a reason for this?”
>     * We believe that the lack of smoothness in some of the plots is due to noise from the random initialization.  Additionally, since we can think of our train and test data as being sampled from an underlying distribution of images, there may be minor fluctuations in the observed trend.  Nevertheless, we both expect and observe the general trend of increasing test loss after a critical depth. Our CNTK experiments (Section 3.3, Appendix D.6), which are deterministic, show that the test loss is monotonically decreasing, and then increasing across a number of settings.
> * “Could the authors elaborate on why a lower Frobenius norm (in the linear networks) is a bad solution?”
>     * In the particular case of this experiment, the minimum Frobenius norm solution is not translation invariant, i.e. the corresponding solution will just place a 1 or -1 in the entry corresponding to the location of the red or blue square in the training set and thus just predicts 0 on the test examples.  In general, the fact that a linear fully connected network will learn the minimum Frobenius-norm solution along with the observation that convolutional networks do better on image classification tasks means that the minimum Frobenius-norm is not necessarily optimal for image classification tasks.  It is indeed possible that convolutional networks minimize the norm in some other space, which is better suited for image classification tasks.
> * “In Fig. 2(e) the test accuracy of the fully-convolutional network actually dips below that of the fully-connected model [...]”
>     * We believe this is due to the effect of random initialization (i.e. not sampling enough random seeds) and randomness in the draw of test data.  However, we feel that understanding this more precisely would be an interesting direction of future work.

---

### Author Response · Authors · 2020-11-25
**Thank you all for the reviews!**

Thank you again for all the helpful feedback regarding our paper.  We wanted to share an overview of the updated experiments below:

1. We have added experiments showing that test error decreases and then increases monotonically with increasing depth (up to 1000 layers) for CNTKs  in section 3.3.  We compare this with the performance of NTKs in this section as well.  We have added in additional experiments for varying numbers of classes in Appendix D.6.

2. We extended the plots for ResNets in Figures 5, 13, 14 and linear CNNs in Figure 7.  We also added in a second linear CNN experiment in Section D.7.

3. We have added in test losses for all of our experiments in Section 3.1 to Appendix D.8.

4. We expanded on our related works section to include  Xiao et al. (2018) and Xiao et al. (2020).

---

### Decision · Program_Chairs · 2021-01-07
**Final Decision**

**Decision:**

Reject

**Comment:**

Reviewers appreciate the numerical results presented in this paper. However, the paper needs a more rigorous theoretical investigation of the empirical phenomenon, or a more comprehensive empirical exploration to pinpoint the key factors. I recommend the authors to incorporate the suggestions from the reviewers and submit the paper to the next top conference.